# Intriguing Properties of Hyperbolic Embeddings in Vision-Language Models

**Sarah Ibrahimi**                                           *s.ibrahimi@uva.nl*
*University of Amsterdam*

**Mina Ghadimi Atigh**                            *m.ghadimiatigh@uva.nl*
*University of Amsterdam*

**Nanne van Noord**                              *n.j.e.vannoord@uva.nl*
*University of Amsterdam*

**Pascal Mettes**                                     *p.s.m.mettes@uva.nl*
*University of Amsterdam*

**Marcel Worring**                                     *m.worring@uva.nl*
*University of Amsterdam*

**Reviewed on OpenReview:** *https://openreview.net/forum?id=P5D2gfi4Gg*

## Abstract

Vision-language models have in short time been established as powerful networks, demonstrating strong performance on a wide range of downstream tasks. A key factor behind their success is the learning of a joint embedding space where pairs of images and textual descriptions are contrastively aligned. Recent work has explored the geometry of the joint embedding space, finding that hyperbolic embeddings provide a compelling alternative to the commonly used Euclidean embeddings. Specifically, hyperbolic embeddings yield improved zero-shot generalization, better visual recognition, and more consistent semantic interpretations. In this paper, we conduct a deeper study into the hyperbolic embeddings and find that they open new doors for vision-language models. In particular, we find that hyperbolic vision-language models provide spatial awareness that Euclidean vision-language models lack, are better capable of dealing with ambiguity, and effectively discriminate between distributions. Our findings shed light on the greater potential of hyperbolic embeddings in large-scale settings, reaching beyond conventional down-stream tasks. Our code is available at `https://github.com/saibr/hypvl`

## 1 Introduction

Continued advances in large-scale contrastive learning between visual data and rich semantic descriptions have produced influential vision-language models, such as Flamingo (Alayrac et al., 2022), FLAVA (Singh et al., 2022), and CLIP (Radford et al., 2021). These models not only allow for multimodal retrieval, their original training objective, but also show to be powerful networks for down-stream zero-shot (Ge et al., 2023b; Menon & Vondrick, 2023; Pratt et al., 2023) and fine-tuning tasks (Luo et al., 2022; Goyal et al., 2023; Paiss et al., 2023). Though commonly optimized in Euclidean space, Desai et al. (2023) recently proposed using hyperbolic embeddings instead, yielding further improvements on retrieval and zero-shot generalization. Since hyperbolic geometry differs vastly from Euclidean geometry, embedding image-text pairs in this alternative space leads to inherently different representational properties. The goal of our work is to uncover the new capabilities that emerge from using a hyperbolic geometry for vision-language models.

The potential of hyperbolic geometry has recently been shown for single-modality analysis in computer vision (Atigh et al., 2021; Ermolov et al., 2022; Franco et al., 2023; Ge et al., 2023b; Liu et al., 2020a; Khrulkov et al., 2020) and language understanding (Tifrea et al., 2019; Zhu et al., 2020; Dhingra et al., 2018). A key advantage of the hyperbolic space is its inherent ability to represent hierarchical structures and complex tree structures with minimal distortion (Ganea et al., 2018a; Nickel & Kiela, 2017; Sala et al., 2018) which is not directly feasible in the Euclidean space (Nickel & Kiela, 2017). Building on those insights, Desai et al. (2023) integrated vision-language models in the hyperbolic space. Their results demonstrate that the benefits generalize to common tasks in the multimodal domain, with competitive zero-shot image classification and improved zero-shot retrieval over Euclidean models, in particular with lower dimensional embeddings. However, an open question remains: are hyperbolic vision-language models useful beyond the common zero-shot classification and image-text retrieval tasks and do hyperbolic embeddings bring new capabilities to vision-language models?

Hyperbolic embeddings have shown advantages over Euclidean embeddings in unimodal settings for tasks including out-of-distribution generalization, uncertainty quantification, and hierarchical representation learning (Mettes et al., 2024). In multimodal scenarios, hyperbolic embeddings might show similar benefits. In this work, we systematically compare Euclidean and hyperbolic image-text embeddings and find three properties where hyperbolic embeddings shine. These properties provide insights into why hyperbolic embeddings achieve strong performance and also demonstrate their potential for real-world impact.

- **Spatial Awareness.** Our analysis reveals that hyperbolic vision-language models have better spatial awareness compared to Euclidean models. While recent benchmarks have been proposed for evaluating spatial reasoning (Zhao et al., 2022; Yüksekgönül et al., 2023; Lewis et al., 2023), large vision-language models still struggle on such tasks, potentially due to contrastive training encouraging shortcut behaviors (Yüksekgönül et al., 2023). We demonstrate that the hyperbolic vision-language model outperforms its Euclidean counterpart on three spatial awareness benchmarks. Notably, the Euclidean model performs no better than random chance on two cases, whereas the hyperbolic model succeeds. To our knowledge, spatial awareness remains unexplored for hyperbolic vision-language works and our findings reveal new potential for hyperbolic embeddings in vision-language learning in overcoming its limitation in spatial reasoning.

- **Ambiguity Resolution.** We find that hyperbolic vision-language models are better in handling ambiguity. While many multimodal tasks involve literal appearances, real-world data contains substantial ambiguity; from words with context-dependent meanings to memes with vague image-text relationships. Ambiguity in a machine learning context can be related to uncertainty and to semantics. In this work, we refer to a specific type of semantic ambiguity. The interpretation of memes and other multimodal polysemic challenges involves a grounding problem where the semantics of the text data can only be resolved by grounding it to the image or video, therefore we will refer to 'visually-grounded language ambiguity' in the remainder of this paper whenever we use the term ambiguity. We demonstrate that the hyperbolic vision-language model outperforms the Euclidean model on three tasks involving multimodal ambiguity: propaganda classification (Dimitrov et al., 2021), visual word sense disambiguation (Raganato et al., 2023) and text-to-GIF retrieval (Song & Soleymani, 2019). A potential explanation is that the hyperbolic space better preserves semantic hierarchies in text embeddings. Through experiments mapping hierarchical relationships, we confirm lower embedding distortion with the hyperbolic model compared to the Euclidean model. We suggest that the improved performance on ambiguity tasks likely comes from implicit hierarchical representations in the hyperbolic embedding space.

- **Out-of-distribution Discrimination.** We find that hyperbolic vision-language models effectively discriminate between in-distribution and out-of-distribution samples. For deployable real-world systems, simply classifying test samples is insufficient and models must detect when inputs differ semantically from the training distribution (Yang et al., 2022). Evaluating across three evaluation metrics, three out-of-distribution detection methods, and five distinct datasets, we consistently observe substantial gains in out-of-distribution detection for the hyperbolic vision-language model compared to the Euclidean model.

With these findings, we shed new light on the importance of the embedding space and the potential of hyperbolic geometry in vision-language models.

## 2 Related Work

### 2.1 Image-Text Representation Learning

Large-scale vision-language models have demonstrated strong performance on a wide variety of downstream tasks (Alayrac et al., 2022; Singh et al., 2022; Radford et al., 2021). The motivation for training such models has been the large availability of images with raw web-text annotations that can be used in training as a way of free-form language supervision. These models have been applied to classification, retrieval and detection across both unimodal and multimodal contexts in zero-shot (Roth et al., 2023; Pratt et al., 2023; Menon & Vondrick, 2023), few-shot (Zhang et al., 2022; P et al., 2023), and full fine-tuning scenarios (Goyal et al., 2023; Ibrahimi et al., 2023; Paiss et al., 2023). Beyond individual tasks, research has also explored robustness to natural distribution shifts (Manli et al., 2022), interpretability (Materzynska et al., 2022) and embedding space properties (Liang et al., 2022). Another emerging area evaluates reasoning about objects, attributes, and relations (Zhao et al., 2022; Yüksekgönül et al., 2023; Lewis et al., 2023). Moreover, recent work from Fang et al. (2022) studied the robustness of CLIP and showed that this is mostly related to the data distribution of the training set, while factors such as the language supervision play a minor part. The dominant performance of vision-language models across a plethora of down-stream tasks has made them a staple technique in the field.

### 2.2 Hyperbolic Learning

Recent years have witnessed rapid progress in hyperbolic learning, as surveyed in Mettes et al. (2024), Peng et al. (2022), and Yang et al. (2023). Specifically in the visual domain, hyperbolic learning has shown to be effective for hierarchical learning (Dhall et al., 2020; Atigh et al., 2021; Long et al., 2020; Liu et al., 2020a), few-shot learning (Gao et al., 2021; Ma et al., 2022; Khrulkov et al., 2020), robust learning (van Spengler et al., 2023; Jie Hong, 2023), and low-dimensional learning (Chami et al., 2020; Nagano et al., 2019; Bose et al., 2020). Beyond these general areas, promising results have been demonstrated for specific vision tasks such as object detection (Lang et al., 2022; Ge et al., 2023a), image segmentation (Atigh et al., 2022), action recognition (Gulshad et al., 2023; Franco et al., 2023), and deep metric learning (Ermolov et al., 2022). For language specific tasks, hyperbolic learning has improved word embeddings (Tifrea et al., 2019; Zhu et al., 2020; Dhingra et al., 2018), graph representations (Chami et al., 2020; Liu et al., 2019) and recommender systems (Mirvakhabova et al., 2020; Wang et al., 2021). The effectiveness of hyperbolic spaces across diverse learning paradigms and tasks emphasizes their potential as a fundamental technique in machine learning.

Desai et al. (2023) introduced the first large-scale hyperbolic vision-language model, demonstrating the emergence of hierarchical structure at text level representations when trained with a loss function enforcing entailment cones (Ganea et al., 2018a). In follow-up work, Shen et al. (2023) evaluated the hierarchical properties of this hyperbolic model on superclass-based image classification and subclass clustering of image embeddings. Their experiments showed comparable performance to a Euclidean baseline, indicating limitations in capturing hierarchical semantics in the image embedding space. Shen et al. (2023) highlighted that hyperbolic vision-language models are not only effective for standard evaluation tasks, but also for interpretability. However, Desai et al. (2023) showed that the hierarchical properties are enforced due to the entailment loss and are therefore most likely present in the text embedding space. In this work, we confirm the hierarchical properties being present in the text embedding space and we show that hyperbolic embeddings in vision-language models indeed open new doors, namely for spatial reasoning, ambiguity, and out-of-distribution discrimination, all out-of-the-box.

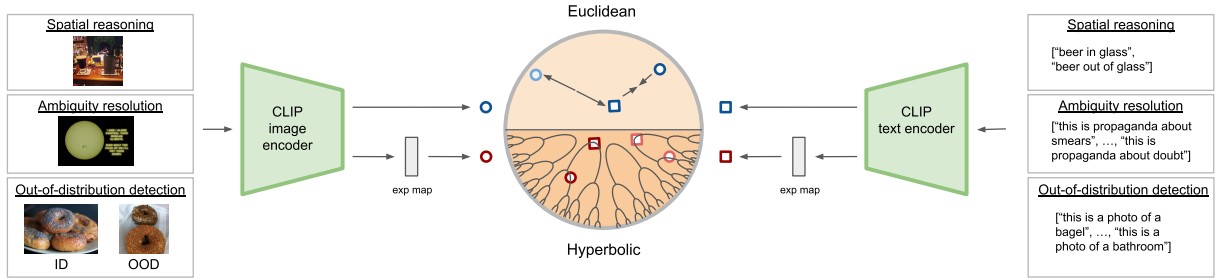

Figure 1: **Hyperbolic and Euclidean Geometry for Vision-Language Representation Learning.** An image and text encoder are used to map images to embeddings in the embedding space. To map the embeddings into the hyperbolic space, an additional exponential mapping function is required, defined by Equation 9. Examples of the visual and textual input for the three properties are presented on the left and right. The color tones indicate whether the embeddings are Euclidean (blue) or hyperbolic (red) and the shapes represent a visual embedding (circle) or a text embedding (square). The Euclidean space brings corresponding image and text embeddings closer together and pushes dissimilar embeddings apart. In the hyperbolic space, the concept of imaginary entailment cones is visualized with a text embedding at the start of the cone and an image embedding within its cone.

## 3 Insights into the Hyperbolic Geometry for Vision-Language Models

### 3.1 Background

**Image-Text Representation Learning**

Typical large-scale vision-language models consist of an image encoder $I$ and a text encoder $T$ that encode the two modalities separately and use a linear projection layer to retrieve the respective embedding. The resulting embeddings $E_I$ and $E_T$ are then $L_2$ normalized and aligned through either (1) a loss function operating on the encoder outputs directly or (2) an additional multimodal encoder. Due to their overwhelming popularity, we focus on (1) with two unimodal encoders and a shared embedding space. Given the image embedding and text embedding, their similarity is measured by the cosine similarity. During training, this similarity is maximized for matching image-text pairs and minimized for dissimilar pairs. The cross-entropy loss is applied over the similarity scores to achieve this. In this paper, we focus on the ubiquitous CLIP model (Radford et al., 2021), which consists of a Vision Transformer (ViT) (Dosovitskiy et al., 2021) as the image encoder and a regular Transformer (Vaswani et al., 2017) as the text encoder following Radford et al. (2019). As CLIP is trained on image-sentence pairs, text prompts are typically formatted into full sentences at inference time to match the training distribution. For the visual recognition task, the common approach is to transform class labels into descriptions akin to "this is a photo of a [class name]" (Menon & Vondrick, 2023; Pratt et al., 2023; Radford et al., 2021). This space is visualized in the upper half of Figure 1.

**Hyperbolic Geometry for Visual and Textual Representation Learning**

Hyperbolic geometry possesses constant negative curvature, contrasting the zero curvature of the flat Euclidean space. This enables unique properties relevant for representation learning, including an inherent hierarchical structure and exponential expansion, meaning that the volume of a ball in hyperbolic space grows exponentially as a function of its diameter, rather than polynomially as in Euclidean geometry. Seminal work on hyperbolic learning leverages these traits for hierarchical embeddings with minimal distortion, vastly outperforming Euclidean embeddings (Nickel & Kiela, 2017). This finding has resulted in a rapid uptake in machine learning, with the development of hyperbolic network layers (Ganea et al., 2018b; Shimizu et al., 2021; van Spengler et al., 2023) and corresponding optimization methods (Lou et al., 2020; Weber et al., 2020).

More generally, hyperbolic space can be represented by multiple models (Cannon et al., 1997), with the Poincaré ball and the Lorentzian hyperboloid as common choices in vision (Mettes et al., 2024) and language

(Peng et al., 2022) representation learning. Each formulation uses a specific distance metric as well as exponential maps, which project points from the tangent (Euclidean) space to hyperbolic space.

Following Desai et al. (2023), we adopt the Lorentzian hyperboloid to avoid numerical instabilities associated with the Poincaré distance metric (Nickel & Kiela, 2018). For completeness, we repeat the definitions for the Lorentzian hyperboloid as presented by Desai et al. (2023) consisting of the Lorentzian inner product, the Lorentzian norm, geodesics, the tangent space, the exponential map and the logarithmic map. Furthermore, we provide the key equations for projecting Euclidean embeddings and measuring similarity in this space, leveraged subsequently in our analysis. The model represents an $n$-dimensional hyperbolic space on the upper half of a two-sheeted hyperboloid in $\mathbb{R}^{n+1}$. Aligning with special relativity theory, Desai et al. (2023) delineate the axis of symmetry as the *time dimension*, with the remaining axes termed *space dimensions*. A vector $\mathbf{x} \in \mathbb{R}^{n+1}$, written as $[\mathbf{x}_{space}, x_{time}]$, comprises space dimensions $\mathbf{x}_{space} \in \mathbb{R}^n$ and a time dimension $x_{time} \in \mathbb{R}$ .

**Lorentzian inner product.** Let $\langle \cdot, \cdot \rangle$ be the Euclidean inner product and $\langle \cdot, \cdot \rangle_{\mathcal{L}}$ the Lorentzian inner product, produced by the Riemannian metric of the Lorentz model. For two vectors $\mathbf{x}, \mathbf{y} \in \mathbb{R}^{n+1}$, the Lorentzian inner product is defined as:

$$\langle \mathbf{x}, \mathbf{y} \rangle_{\mathcal{L}} = \langle \mathbf{x}_{space}, \mathbf{y}_{space} \rangle - x_{time}\, y_{time} \tag{1}$$

**Lorentzian norm.** This is defined by $\|\mathbf{x}\|_{\mathcal{L}} = \sqrt{|\langle \mathbf{x}, \mathbf{x} \rangle_{\mathcal{L}}|}$. The Lorentz model having a constant curvature $-c$ is defined as the set of vectors:

$$\mathcal{L}^n = \{\mathbf{x} \in \mathbb{R}^{n+1} \mid \langle \mathbf{x}, \mathbf{x} \rangle_{\mathcal{L}} = {}^{-1}/c\}\, ,\ c > 0 \tag{2}$$

All vectors in this set satisfy the following constraint:

$$x_{time} = \sqrt{{}^1/c + \|\mathbf{x}_{space}\|^2} \tag{3}$$

**Geodesics.** A geodesic arc is the distance minimizing curve between two points on the manifold. Geodesics in the Lorentz model are intersections of the hyperboloid with hyperplanes passing through the origin of $\mathbb{R}^{n+1}$. The Lorentzian distance between two points $\mathbf{x}, \mathbf{y} \in \mathcal{L}^n$ is:

$$d_{\mathcal{L}}(\mathbf{x}, \mathbf{y}) = \sqrt{{}^1/c} \cdot \cosh^{-1}(-c\, \langle \mathbf{x}, \mathbf{y} \rangle_{\mathcal{L}}) \tag{4}$$

**Tangent space.** The tangent space at a point $\mathbf{z} \in \mathcal{L}^n$ is a Euclidean space of vectors that are orthogonal to $\mathbf{z}$ according to the Lorentzian inner product:

$$\mathcal{T}_{\mathbf{z}}\mathcal{L}^n = \{\mathbf{v} \in \mathbb{R}^{n+1} \mid \langle \mathbf{z}, \mathbf{v} \rangle_{\mathcal{L}} = 0\} \tag{5}$$

An orthogonal projection can project a vector in ambient space $\mathbf{u} \in \mathbb{R}^{n+1}$ to the tangent space $\mathcal{T}_{\mathbf{z}}\mathcal{L}^n$.

$$\mathbf{v} = \text{proj}_{\mathbf{z}}(\mathbf{u}) = \mathbf{u} + c\, \mathbf{z}\, \langle \mathbf{z}, \mathbf{u} \rangle_{\mathcal{L}} \tag{6}$$

**Exponential map.** This maps a vector $\mathbf{v}$ from tangent spaces onto $\mathbf{x}$ on the manifold. For a point $\mathbf{z}$ on the hyperboloid, it is defined as $\text{expm}_{\mathbf{z}} : \mathcal{T}_{\mathbf{z}}\mathcal{L}^n \to \mathcal{L}^n$ with the expression:

$$\mathbf{x} = \text{expm}_{\mathbf{z}}(\mathbf{v}) = \cosh(\sqrt{c}\, \|\mathbf{v}\|_{\mathcal{L}})\, \mathbf{z} + \frac{\sinh(\sqrt{c}\, \|\mathbf{v}\|_{\mathcal{L}})}{\sqrt{c}\, \|\mathbf{v}\|_{\mathcal{L}}}\, \mathbf{v} \tag{7}$$

**Logarithmic map.** This is the reverse of the exponential map ($\text{logm}_{\mathbf{z}} : \mathcal{L}^n \to \mathcal{T}_{\mathbf{z}}\mathcal{L}^n$), that maps $\mathbf{x}$ from the hyperboloid back to $\mathbf{v}$ in the tangent space:

$$\mathbf{v} = \text{logm}_{\mathbf{z}}(\mathbf{x}) = \frac{\cosh^{-1}(-c\, \langle \mathbf{z}, \mathbf{x} \rangle_{\mathcal{L}})}{\sqrt{(c\, \langle \mathbf{z}, \mathbf{x} \rangle_{\mathcal{L}})^2 - 1}}\, \text{proj}_{\mathbf{z}}(\mathbf{x}) \tag{8}$$

We follow Desai et al. (2023) and only consider the maps where $\mathbf{z}$ is the origin of the hyperboloid ($\mathbf{O} = [\mathbf{0}, \sqrt{1/c}]$).

**Projection of embeddings.** We project the Euclidean image and text embeddings $\mathbf{E_I}$ and $\mathbf{E_T}$ into the Lorentzian manifold via the exponential mapping. Since $\mathbf{E_I}, \mathbf{E_T} \in \mathbb{R}^n$, a transformation needs to be applied to make sure that the projected embeddings lie on the Lorentz hyperboloid $\mathcal{L}^n$ in $\mathbb{R}^{n+1}$. This can be done by $\mathbf{E_I}^+ = [\mathbf{E_I}, 0] \in \mathbb{R}^{n+1}$ and $\mathbf{E_T}^+ = [\mathbf{E_T}, 0] \in \mathbb{R}^{n+1}$. Embeddings belong to the tangent space at the hyperboloid origin $\mathbf{O}$ and therefore only the space dimensions are used in this projection, which results in $\mathbf{E_I} = \mathbf{E_{I_{space}}}$ and $\mathbf{E_T} = \mathbf{E_{T_{space}}}$. Now Equation 7 can be simplified to the following equation for $k \in \{\mathbf{I}, \mathbf{T}\}$:

$$\mathbf{E}^*_{k_{space}} = \frac{sinh(\sqrt{c}\, \|\mathbf{E}_{k_{space}}\|)}{\sqrt{c}\, \|\mathbf{E}_{k_{space}}\|}\mathbf{E}_{k_{space}}, \quad k \in \{\mathbf{I}, \mathbf{T}\} \tag{9}$$

We measure the similarity between $\mathbf{E_I^*}$ and $\mathbf{E_T^*}$ using the Lorentzian inner product from Equation 1. For this calculation, we need the *time* components that can be calculated using Equation 3 for $\mathbf{E}^*_{I_{space}}$ and $\mathbf{E}^*_{T_{space}}$. The inner product results in:

$$\langle \mathbf{E_I^*}, \mathbf{E_T^*} \rangle_{\mathcal{L}} = \langle \mathbf{E^*_{I_{space}}}, \mathbf{E^*_{T_{space}}} \rangle - E^*_{I_{time}}\, E^*_{T_{time}} \tag{10}$$

Desai et al. (2023) train a hyperbolic CLIP model using a combination of contrastive and entailment losses. The contrastive component mirrors the loss from Radford et al. (2021), substituting the Euclidean similarity function with the negative Lorentzian distance function. The entailment loss is based on work by Ganea et al. (2018a) and Le et al. (2019) and aims to enforce hierarchical structure. Specifically, image embeddings $\mathbf{E_I^*}$ are pushed into imaginary cones projected by the corresponding text embeddings $\mathbf{E_T^*}$, positioning visual concepts towards the edges of the hyperbolic space. This space is visualized in the lower half of Figure 1. Since fine-tuning and implementation details for these losses are out of scope of this work, we refer readers to Desai et al. (2023) for specifics.

Next, we discuss three assumptions on the potential benefits of hyperbolic geometry for vision-language modeling.

### 3.2 Assumption 1: Hyperbolic vision-language models exhibit the potential for spatial awareness.

Recent work has analyzed the spatial reasoning capacities of pre-trained Euclidean vision-language models by studying compositional relationships, regarding objects, attributes and relations (Yüksekgönül et al., 2023; Zhao et al., 2022; Lewis et al., 2023). Results show poor zero-shot performance, with models failing to capture semantic relationships between objects, attributes, and relations (Yüksekgönül et al., 2023; Zhao et al., 2022; Lewis et al., 2023). The models exhibit "bag-of-words" behavior indicated by Yüksekgönül et al. (2023) and Zhao et al. (2022), with performance unrelated to scale or training data size. Yüksekgönül et al. (2023) hypothesize that contrastive pre-training enables shortcuts that inhibit learning true compositionality. Unlike Euclidean alternatives, hyperbolic CLIP uses an additional entailment loss that may prevent shortcuts and enforce latent hierarchies during training. We expect this geometric structure and optimization offers improved spatial awareness lacking in conventional vision-language models.

### 3.3 Assumption 2: Hyperbolic vision-language models are better in dealing with ambiguity.

Ambiguity is pervasive in language (Piantadosi et al., 2012). More important, ambiguity is omnipresent in real-world multimodal data, making it an important aspect to consider for deployment. As Chen et al. (2022) observed, ambiguity adds representational granularity by presenting multiple perspectives on identical concepts. We expect that hyperbolic vision-language models are better equipped to tackle ambiguity given their latent hierarchical organization. We investigate this assumption by first quantitatively comparing Euclidean and hyperbolic models on tasks with ambiguity, after which we dive deeper into the hierarchical nature of both models.

### 3.4 Assumption 3: Hyperbolic vision-language models are good out-of-distribution discriminators.

Hyperbolic learning has demonstrated potential for robustness tasks and uncertainty quantification in uni-modal scenarios (Mettes et al., 2024). A critical challenge in this domain is out-of-distribution (OOD) detection, aiming to identify inputs that deviate from the distribution of training or fine-tuning data. OOD detection links to several active research areas like open-set recognition, anomaly detection, novelty detection, and selective prediction (Yang et al., 2022). We expect that the geometric properties of hyperbolic embeddings can enhance distributional awareness in vision-language models. Specifically, the concentration of semantic concepts combined with expansion towards the borders could sharpen discrimination of in- versus out-of-distribution data. We test this via OOD detection tasks comparing hyperbolic and Euclidean CLIP.

### 3.5 Setup

Throughout this paper, we compare two models, namely the Euclidean CLIP model and the hyperbolic CLIP model presented by Desai et al. (2023). In their work, they trained ViT-S/16, ViT-B/16 and ViT-L/16 backbones and they publicly released their model checkpoints, providing one checkpoint per model. Both Euclidean and hyperbolic models share the same encoder architecture and were trained on 12M image-text pairs from RedCaps (Desai et al., 2021).

In our study, we evaluate the Euclidean and hyperbolic ViT-L/16 backbones on all tasks and for each property, we add a comparison study including all backbone sizes. For fair assessment, we use the provided checkpoints from Desai et al. (2023) in a zero-shot evaluation mode. Thus, we do not train nor finetune any model ourselves. All models output 512-dim embeddings, we will use these embeddings directly for all our experiments and do not add any additional linear layers to the model.

Since we have only one checkpoint for each model, we are unable to average the results over multiple checkpoints. However, we provide statistical significance tests on the results by using a bootstrapping technique on the test set. More specifically, we follow Sanchez-Lengeling et al. (2019) by using a resampling method with replacement on the test set for 500 times, due to common practice. This results in a distribution of 500 data points for which we report the mean and standard deviation for all experiments. We perform one-sided t-tests on the data distributions of the Euclidean and hyperbolic models with the null hypothesis stating that the *data distribution of the hyperbolic model has a higher mean value than the Euclidean model* with a significance level of 2.5%.

Tasks use the original similarity score from CLIP (Radford et al., 2021), which is the cosine similarity, and the Lorentzian inner product for hyperbolic CLIP, unless otherwise specified. An overview of the pipeline is presented in Figure 1.

## 4 Results

### 4.1 Hyperbolic vision-language models exhibit spatial awareness

**Setup.** To study the first property, we take three benchmarks that have been recently formulated to study spatial awareness: *VL-Checklist* (Zhao et al., 2022), *VG-Relations* (Yüksekgönül et al., 2023) and *CLIPbind-r* (Lewis et al., 2023). In all datasets, the setup is a relation $R$ between two nouns $a$ and $b$, with notation $aRb$.

- **VL-Checklist**: a spatial reasoning benchmark dataset that uses 30k Visual Genome (Krishna et al., 2017) images with two different descriptions, a correct description $d_+$ with relation $R$ and a hard-negative description $d_-$ with a different relation $S$ ($aSb$). The model is given an image with two sentences as input and returns similarity scores between the image and both sentences, the Lorentzian inner products $\langle \mathbf{E}_\mathbf{I}^*, \mathbf{E}_{\mathbf{T}d_+}^* \rangle_\mathcal{L}$ and $\langle \mathbf{E}_\mathbf{I}^*, \mathbf{E}_{\mathbf{T}d_-}^* \rangle_\mathcal{L}$ for the hyperbolic model and cosine-similarity between $\mathbf{E}_\mathbf{I}$ and $\mathbf{E}_{\mathbf{T}d_+}$ and between $\mathbf{E}_\mathbf{I}$ and $\mathbf{E}_{\mathbf{T}d_-}$ for the Euclidean model. When $\langle \mathbf{E}_\mathbf{I}^*, \mathbf{E}_{\mathbf{T}d_+}^* \rangle_\mathcal{L} > \langle \mathbf{E}_\mathbf{I}^*, \mathbf{E}_{\mathbf{T}d_-}^* \rangle_\mathcal{L}$, the prediction is considered as correct for the hyperbolic model. The same holds for the cosine similarity scores. Accuracy is the percentage of images correctly scored higher for the true relation. As this is a binary task, random chance is 50%.

Table 1: **Hyperbolic vision-language models exhibit the potential for spatial awareness.** The Euclidean CLIP exhibits little to no spatial awareness for *VL-Checklist*, even leading to results worse than random guessing for some spatial tasks (*CLIPbind-r*). Hyperbolic CLIP performs clearly better across all three challenging spatial tasks, highlighting that these models do not shortcut spatial relations between objects and attributes and provide a richer embedding space, even though they are trained on the exact same data. L, B, and S refer to the ViT-L/16, ViT-B/16 and ViT-S/16 models and for all results indicated with ‡ the hyperbolic model is statistically significant better than the Euclidean model with $p < 0.025$.

| Model | VL-Checklist | | | VG-Relations | | | CLIPbind-r | | |
|---|---|---|---|---|---|---|---|---|---|
| | S | B | L | S | B | L | S | B | L |
| Random guess | 50.0 | 50.0 | 50.0 | 50.0 | 50.0 | 50.0 | 20.0 | 20.0 | 20.0 |
| Euclidean CLIP | $55.3 \pm 1.6$ | $52.1 \pm 1.5$ | $52.3 \pm 1.6$ | $58.5 \pm 1.7$ | $57.9 \pm 1.6$ | $56.3 \pm 1.7$ | $16.6 \pm 0.2$ | $4.6 \pm 0.1$ | $12.8 \pm 0.2$ |
| Hyperbolic CLIP | $54.8 \pm 1.6$ | $57.6 \pm 1.5^{\ddagger}$ | $\mathbf{59.9 \pm 1.5^{\ddagger}}$ | $\mathbf{61.5 \pm 1.8^{\ddagger}}$ | $59.2 \pm 1.5^{\ddagger}$ | $58.9 \pm 1.7^{\ddagger}$ | $12.6 \pm 0.2$ | $17.0 \pm 0.2^{\ddagger}$ | $\mathbf{25.2 \pm 0.2^{\ddagger}}$ |

- **VG-Relations**: a spatial reasoning benchmark dataset that consists of 48 relations with nearly 24k test cases, with images from Visual Genome (Krishna et al., 2017), containing both spatial relations such as "in front of" and "below" as well as actions such as "eating" and "standing on". Each image is paired with two descriptions, one being correct and one incorrect, similar to *VL-Checklist*. However, the main difference is that instead of a change in relation, the order of the two nouns is reversed, which results in $aRb$ and $bRa$. The score calculations are the same as for *VL-Checklist*.

- **CLIPbind-r**: a synthetic CLEVR-inspired benchmark dataset (Johnson et al., 2017) that is created by a tool for 3D modelling and rendering (Lewis et al., 2023) with 10k images for validation. Each image has two objects, chosen out of a set of three, and one relation, out of four possibilities, which results in 24 combinations of spatial relations. Next to the pair $aRb$, four hard negatives are created with a different relation $S$, a different object $c$ and reverse ordering, namely $d_- = \{bRa, aSb, aRc, cRb\}$. Similar to the other tasks, this task looks at the highest similarity scores between the image and the spatial relation compositions, except that there are four distractors. With five candidates per image, random performance is 20%.

**Results.** Quantitative results in Table 1 show clear advantages for the hyperbolic model on the three benchmarks for the best-performing model size. On *VL-Checklist*, the large model achieves the highest performance and for this model size the Euclidean CLIP scores near random chance (50%), while the hyperbolic architecture achieves 59.9% accuracy which is significantly higher. For *CLIPbind-r* the Euclidean model scores lower than random for all model sizes, which is similar to the behavior of some models evaluated in Lewis et al. (2023), but we notice that the hyperbolic model with the large size performs better than random. In fact, the CLIP trained by OpenAI (Radford et al., 2021), which has been pre-trained on 400M image-text pairs, has an accuracy of 26.8%, which is only slightly higher than the hyperbolic CLIP trained on 12M image-text pairs. The gap reduces on *VG-Relations* but remains substantial at over 3% absolute improvement. For *VL-Checklist* and *CLIPbind-r*, the large hyperbolic model achieves the highest scores, where for *VG-Relations* the small model performs best. We can conclude that overall the large hyperbolic model provides the most consistent results, since it does perform better than random on two tasks.

**Analysis.** To gain further insights into which spatial properties are better preserved in hyperbolic vision-language models, we have performed additional qualitative and quantitative analyses. In Figure 2, we show a few samples from the *VL-Checklist* dataset, indicating the correct/incorrect relation by color. Euclidean CLIP struggles with more complex relations such as "on"/"under" and in "between"/"nearby" compared to the hyperbolic alternative. For spatial relations that are contextual, such as "near"/"far from", both models still struggle, as such cases require explicit knowledge about the real-world size of the corresponding objects.

Since the *VG-Relations* benchmark uses spatial prepositions and action verbs, we analyze performance per relation in Figure 3, focusing on frequently occurring spatial terms. For this set, we use the relations and actions with spatial relations with at least 20 occurrences in the dataset. For a few relations such as "at", "above", "sitting at", and "lying on", Euclidean CLIP is able to compete with hyperbolic CLIP. For many other relations however, large differences occur, especially for "standing on", "sitting on", and "looking at".

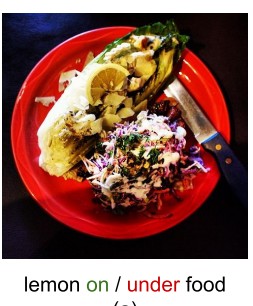 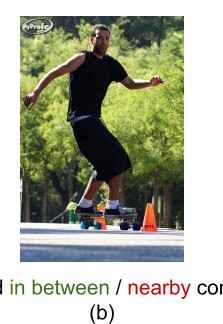 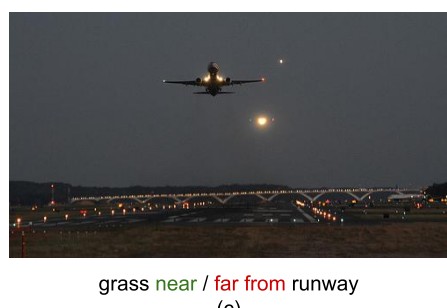

lemon on / under food      road in between / nearby cone      grass near / far from runway
(a)                  (b)                     (c)

Figure 2: **Success and failure cases of hyperbolic spatial reasoning.** We show three examples from the *VL-Checklist* spatial split, each with an image and two relational descriptions: one correct (green) and one incorrect (red), *e.g.*, lemon on food (correct) and lemon under food (incorrect) in example (a). For samples (a) and (b) the hyperbolic model assigns higher similarity to the correct relations, while the Euclidean embeddings score the incorrect relation higher. However, in (c) both models fail because of incorrectly preferring "grass far from runway" over the true "grass near runway" description, likely due to the high contextual nature of the relation as it requires explicit knowledge about the size of the object and understanding about the distance between the photographer and the objects in the scene.

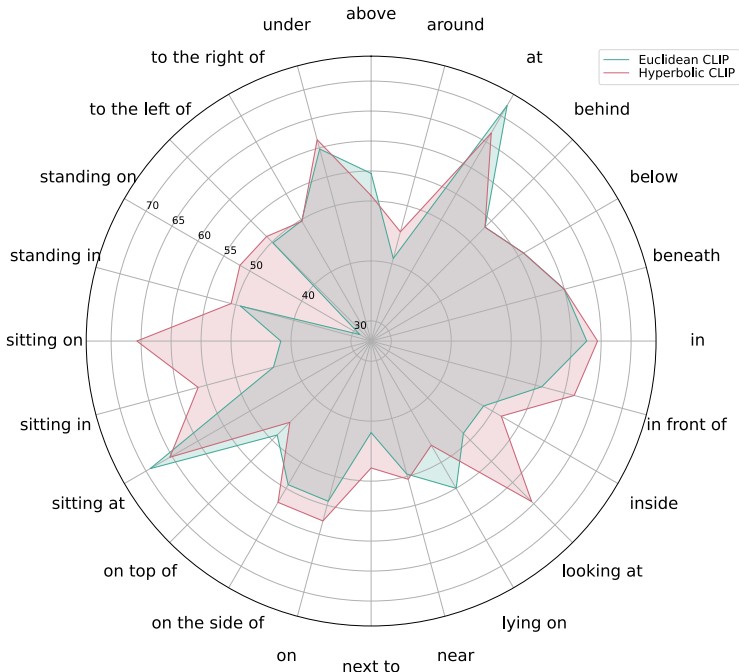

Figure 3: **Hyperbolic CLIP covers more spatial relations.** In this plot, the difference in performance is shown for spatial relations with more than 20 samples. Hyperbolic and Euclidean CLIP are competitive for a number of spatial relations, but hyperbolic CLIP performs much better for others, highlighting a more robust and diverse spatial awareness in hyperbolic CLIP.

Overall, hyperbolic CLIP is more robust to the different spatial relations and provides a better option across all settings.

Models that perform well on spatial awareness tasks are useful for many applications, since it requires an understanding of order and compositionality. For real-world examples, this could be *e.g.*, systems assisting visually impaired people or robots in their interactions with people or in industrial environments.

### 4.2 Hyperbolic vision-language models are better in dealing with ambiguity

**Setup.** Where deep networks excel when given clearly defined semantic labels, they struggle to deal with ambiguity (Kim et al., 2023). Ambiguity can take many forms, from disambiguating words to dealing with data that is ambiguous and playful by design, such as memes. Here we investigate which embedding space is more suited for dealing with ambiguity. We study three benchmarks:

- **Propaganda Memes**: This benchmark dataset contains 950 ambiguous image-text memes spanning 22 propaganda techniques, including whataboutism, obfuscation, and glittering generalities (Dimitrov et al., 2021). Each meme comes with an extracted textual description and one or more category labels. The task is multi-label classification with models ranking similarity between prompted label embeddings of the form "this is propaganda about [class label]" and the meme embedding. For the hyperbolic model, we compute the similarity score by the Lorentzian inner product given by Equation 1 between the meme embedding and the text embedding of the prompted class names $\langle \mathbf{E}^*_{\mathbf{meme}}, \mathbf{E}^*_{\mathbf{T}} \rangle_{\mathcal{L}}$, with three variants for the meme embedding:
    1. text-only $\mathbf{E}^*_{\mathbf{meme}} = \mathbf{E}^*_{\mathbf{meme}_{\text{text}}}$
    2. image-only $\mathbf{E}^*_{\mathbf{meme}} = \mathbf{E}^*_{\mathbf{meme}_{\text{image}}}$
    3. averaged embeddings $\mathbf{E}^*_{\mathbf{meme}} = (\mathbf{E}^*_{\mathbf{meme}_{\text{text}}} + \mathbf{E}^*_{\mathbf{meme}_{\text{image}}})/2$

    A fourth setup is analyzed where the scores of the text-only and image-only embeddings are averaged:
    4. averaged similarity scores $(\langle \mathbf{E}^*_{\mathbf{meme}_{\text{image}}}, \mathbf{E}^*_{\mathbf{T}} \rangle_{\mathcal{L}} + \langle \mathbf{E}^*_{\mathbf{meme}_{\text{text}}}, \mathbf{E}^*_{\mathbf{T}} \rangle_{\mathcal{L}})/2$.

    The Euclidean model uses the cosine similarity on the same variations. We evaluate with the weighted mean Average Precision described in (Wang et al., 2023).

- **My Reaction When**: This benchmark dataset has 50K video-sentence pairs from social media depicting physical/emotional reactions to textual captions including a high ambiguity (Song & Soleymani, 2019). Videos are GIFs representing responses to the paired sentences. We evaluate on video-text and text-video retrieval using widely used mean pooling to aggregate per-frame CLIP embeddings into a single video representation (Luo et al., 2022). For hyperbolic CLIP, video-text similarity is the Lorentzian inner product between mean pooled video and sentence embeddings, $\langle \mathbf{E}^*_{\mathbf{video}}, \mathbf{E}^*_{\mathbf{T}} \rangle_{\mathcal{L}}$. Euclidean CLIP uses cosine similarity. Evaluation is via recall@k (top-k accuracy), checking if ground truth pairs are retrieved in closest neighborhoods. This tests semantic matching between ambiguous pairs of reactions and descriptive captions.

- **Visual Word Sense Disambiguation**: This benchmark evaluates associating ambiguous words like "mouse", with intended meanings from contextual images (Raganato et al., 2023). Each case has a target word and 10 candidate images, one of which matches its sense. Words are provided alone or with 1-2 word phrases for minimal context, e.g. "andromeda" alone vs "andromeda tree". Models rank image similarity to the word/phrase to identify the intended visual depiction among distractors. We compute hyperbolic CLIP's Lorentzian inner product and Euclidean CLIP's cosine similarity, following the setup in (Raganato et al., 2023). We evaluate the model on the full dataset that consists of 13332 samples. Evaluation uses Mean Reciprocal Rank at 5 and 10 (MRR@5, MRR@10) plus Hit Rate at 1 (HIT@1) to measure disambiguation capabilities given varying levels of contextual grounding.

The *Propaganda Memes* and *My Reaction When* benchmark datasets involve ambiguity in aligning textual captions with corresponding visuals. Memes combine images and text for rhetorical effect, with complex connotations that obscure intended meanings. Likewise, associating social media posts to emotive animated GIFs relies on interpreting nuanced contextual implications. On the contrary, the *Visual Word Sense Disambiguation* benchmark directly evaluates resolution of ambiguous words through selection among depictive image candidates. Rather than complex compositional understanding between modalities, the core challenge is isolating the intended sense of polysemous concepts given limited grounding context.

While ambiguity types vary, all tasks require models to handle uncertainty in cross-modal correlations: sociopolitical, affective, or lexical. They measure how well vision-language models leverage context to illuminate intended meanings amidst ambiguity spanning text, static visuals, and video.

Table 2: **Hyperbolic vision-language models can deal better with ambiguity (I)**, as demonstrated on the *Propaganda memes* benchmark. Regardless of whether we investigate text only, image only, or averaging image-text features or predictions, hyperbolic embeddings provide vastly better results. The results are for the ViT-L/16 backbone and for all settings the improvement by the hyperbolic model is statistically significant.

| Model | Image only | Text only | Avg image-text feats | Avg image-text preds |
|---|---|---|---|---|
| Euclidean CLIP | 36.3 ± 0.9 | 33.5 ± 0.8 | 33.9 ± 0.8 | 33.9 ± 0.8 |
| Hyperbolic CLIP | **37.7 ± 0.9**[‡] | **34.2 ± 0.8**[‡] | **35.9 ± 0.9**[‡] | **36.0 ± 0.9**[‡] |

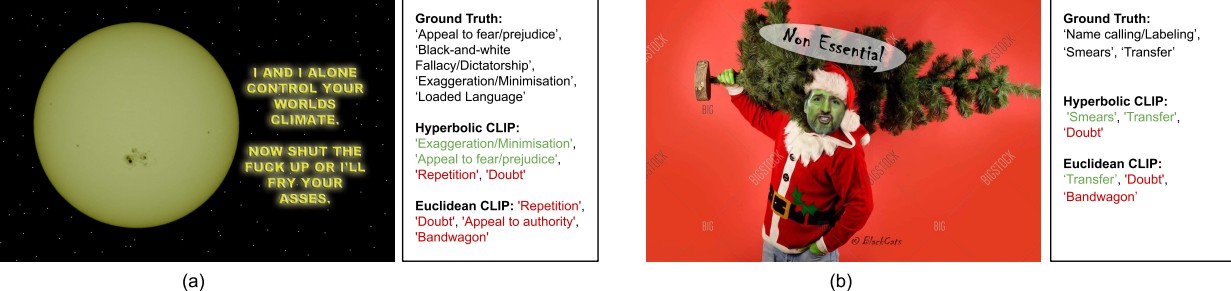

(a)                                                                              (b)

Figure 4: **Propaganda Memes samples.** We show two examples from the Propaganda Memes benchmark dataset with their ground truth labels and the predictions of both models where the meme embedding is the average of the image and text embedding. In example (a), the hyperbolic CLIP model is able to predict two out of four ground truth labels correctly whereas the Euclidean model does not predict any of those. In example (b), the hyperbolic is able to predict two out of three categories correctly, including smears, where its Euclidean counterpart has only one correct prediction.

**Results.** In Table 2, we show the results on the *Propaganda Memes* benchmark. We see that the hyperbolic embeddings perform significantly better than their Euclidean counterpart. Dimitrov et al. (2021) have indicated that smears, loaded language and name calling/labeling are the most common meme types in propaganda memes, with smears covering more than half of the dataset, where they argue that this dataset is a good representation of memes that can be found on social media. They define smears as an effort to damage or call into question someone's reputation, by propounding negative propaganda. Examples of memes are given in Figure 4.

Table 3 shows text-video retrieval results on the *My Reaction When* validation (1K) and test (5K) sets. Scaling up the dataset increases the retrieval difficulty, as observed from the decreased performance on the test set. Still, hyperbolic CLIP consistently surpasses its Euclidean variant, particularly for top few (small $k$) retrieved results.

Table 4 shows *Visual Word Sense Disambiguation* results, with hyperbolic CLIP outperforming Euclidean on both phrase and word inputs. Notably, gains persist even when only the ambiguous word is provided without any disambiguating context. This suggests inherent benefits in reasoning about polysemous lexical ambiguity. However, strong overall performance remains elusive - vision-language models struggle to resolve fine-grained distinctions absent in pre-training data like RedCaps. In summary, while hyperbolic space better handles ambiguity, truly contending with granular real-world ambiguity requires acquiring specialist visual knowledge of distinctions like mouse-animal versus mouse-computer. Nevertheless, the consistent improvements validate geometric advantages in multimodal ambiguity.

**Analysis.** One potential explanation for the superior performance of hyperbolic CLIP on ambiguity tasks is its inherent hierarchical embedding space. Ambiguity introduces additional granularity in recognition tasks by adding an instance-level layer (Chen et al., 2022). However, hyperbolic vision-language models are trained on the same image-text pairs as Euclidean models rather than explicit hierarchies. Recent work showed limitations in visual hierarchy encoding (Shen et al., 2023), but the textual encoder capturing task semantics is more relevant here. Before relating ambiguity and hierarchy, we first verify if hyperbolic text embeddings exhibit stronger latent hierarchical properties.

Table 3: **Hyperbolic vision-language models can deal better with ambiguity (II)**, as demonstrated on the *My Reaction When* benchmark. Also when dealing with video inputs and emotional ambiguity, hyperbolic embeddings are preferred. These results are for the ViT-L/16 backbone. Most of the results are statistically significant except for the r@1 for text-video retrieval on the validation set and r@1 and r@5 for video-text retrieval on the test set. Note that the results are reported in one decimal due to space constraints of the table. The r@1 for text-video retrieval on the test-set does not have the exact same decimals but is in fact $0.100 \pm 0.044$ for the Euclidean model and $0.120 \pm 0.049$ for the hyperbolic model and with that statistically significant with $p < 0.025$, indicated with ‡.

| | Val-set(1k) | | | | | | | |
| Model | Text-Video Retrieval | | | | Video-Text Retrieval | | | |
| | r@1 | r@5 | r@10 | r@20 | r@1 | r@5 | r@10 | r@20 |
| Euclidean CLIP | **0.2 ± 0.1** | 0.8 ± 0.3 | 1.7 ± 0.4 | 3.3 ± 0.5 | 0.2 ± 0.1 | 0.6 ± 0.2 | 1.6 ± 0.4 | 2.9 ± 0.5 |
| Hyperbolic CLIP | **0.2 ± 0.1** | **1.3 ± 0.4‡** | **2.4 ± 0.5‡** | **4.2 ± 0.6‡** | **0.5 ± 0.2‡** | **1.2 ± 0.4‡** | **2.4 ± 0.5‡** | **4.2 ± 0.6‡** |
| | Test-set(5k) | | | | | | | |
| | r@1 | r@5 | r@10 | r@20 | r@1 | r@5 | r@10 | r@20 |
| Euclidean CLIP | **0.1 ± 0.0** | 0.3 ± 0.1 | 0.5 ± 0.1 | 0.9 ± 0.1 | **0.1 ± 0.1** | **0.5 ± 0.1** | 0.7 ± 0.1 | 1.2 ± 0.1 |
| Hyperbolic CLIP | **0.1 ± 0.0‡** | **0.5 ± 0.1‡** | **1.0 ± 0.1‡** | **1.4 ± 0.2‡** | **0.1 ± 0.1** | **0.5 ± 0.1** | **0.9 ± 0.1‡** | **1.5 ± 0.2‡** |

Table 4: **Hyperbolic vision-language models can deal better with ambiguity (III)**, as demonstrated on the *Visual Word Sense Disambiguation* benchmark. The hyperbolic CLIP outperforms its Euclidean variant in ranking images similar to the ambiguous word or phrase. L, B, and S refer to the ViT-L/16, ViT-B/16 and ViT-S/16 models and for all settings the improvement by the hyperbolic model is statistically significant, indicated with ‡. The large backbone performs best for all setups and metrics.

| Model | Target | HIT | | | MRR@5 | | | MRR@10 | | |
| | | S | B | L | S | B | L | S | B | L |
| Euclidean CLIP | phrase | 47.7 ± 0.4 | 49.2 ± 0.5 | 49.1 ± 0.5 | 61.2 ± 0.3 | 62.4 ± 0.4 | 62.3 ± 0.3 | 63.3 ± 0.3 | 64.4 ± 0.3 | 64.3 ± 0.3 |
| Hyperbolic CLIP | phrase | 48.3 ± 0.5‡ | 49.3 ± 0.4‡ | **49.4 ± 0.4‡** | 61.6 ± 0.4‡ | 62.5 ± 0.3‡ | **62.6 ± 0.3‡** | 63.7 ± 0.3‡ | 64.5 ± 0.3‡ | **64.6 ± 0.3‡** |
| | | S | B | L | S | B | L | S | B | L |
| Euclidean CLIP | word | 39.8 ± 0.4 | 40.8 ± 0.4 | 40.9 ± 0.4 | 53.4 ± 0.4 | 54.1 ± 0.3 | 54.0 ± 0.3 | 56.3 ± 0.3 | 57.0 ± 0.3 | 57.0 ± 0.3 |
| Hyperbolic CLIP | word | 41.1 ± 0.4‡ | 41.5 ± 0.4‡ | **41.9 ± 0.4‡** | 54.1 ± 0.4‡ | 54.9 ± 0.4‡ | **55.2 ± 0.4‡** | 57.1 ± 0.3‡ | 57.7 ± 0.3‡ | **58.0 ± 0.3‡** |

To quantify textual hierarchy encoding, we leverage label trees from six vision datasets: CIFAR-100, Animals with Attributes 2 (AWA2) (Xian et al., 2019), PASCAL-VOC (Everingham et al., 2010), UCF (Soomro et al., 2012), Kinetics (Kay et al., 2017), and ActivityNet (Heilbron et al., 2015). The first three are provided by the creators of the datasets, while the latter three are taken from Long et al. (2020) and Gulshad et al. (2023). Leaf nodes are encoded as "this is a photo of a [class name]" sentences using Euclidean and hyperbolic CLIP text encoders. We evaluate whether hierarchy distance corresponds with embedding distance, where hierarchy distance between classes is defined by tree levels to the common ancestor. We use the Pearson correlation coefficient to measure whether a linearity between the embedding distances and hierarchy distances is present.

Table 5 shows Pearson correlation between embedding and hierarchy distances. Hyperbolic CLIP exhibits higher correlation than Euclidean variants on five of six datasets, demonstrating improved latent encoding of hierarchical relationships. To validate that these gains span beyond the lowest hierarchy level, we filter leaf nodes from the ActivityNet taxonomy and remeasure correlations. Even within higher layers, hyperbolic embeddings show greater alignment with tree distances (Euclidean: 0.2434, Hyperbolic: 0.2823).

Models that are able to deal with unclear and subjective situations are required in practice. It requires models to learn how to incorporate contexts into the decision-making process. A practical application for this task could be monitoring of posts of users on online platforms. A model should be able to interpret all sorts of memes and other posts based on the visible information in the post itself and well-known contextual information. This task is still even difficult for humans and therefore far from solvable for models, but these benchmarks are a required initial step.

In summary, hyperbolic text encoders intrinsically structure semantic concepts more hierarchically, which is an advantage originating from the underlying geometry. We hypothesize these latent textual hierarchies aid in resolving ambiguity by providing a framework to structurally represent uncertain, fine-grained distinctions.

Table 5: **Hyperbolic vision-language models are more hierarchically structured.** We embed six hierarchies in Euclidean and hyperbolic embeddings through prompting and find that hyperbolic embeddings better correlate with the underlying structure of the hierarchies. While trained on exactly the same data, the hierarchical nature of hyperbolic geometry allows for a better latent hierarchical structuring of semantics.

| Model | PASCAL-VOC | AWA2 | UCF101 | CIFAR-100 | Kinetics | ActivityNet |
|---|---|---|---|---|---|---|
| Hierarchy levels | 4 | 6 | 3 | 3 | 3 | 3 |
| Number of concepts | 55 | 77 | 125 | 127 | 239 | 244 |
| Euclidean CLIP | 0.3689 | 0.2313 | **0.1200** | 0.2683 | 0.1675 | 0.1450 |
| Hyperbolic CLIP | **0.4562** | **0.2861** | 0.1179 | **0.3646** | **0.2190** | **0.1830** |

However, directly confirming the connection between hierarchy and improved ambiguity handling remains an open question for future work and would require extensive further analysis into geometric mechanisms granting these advantages.

### 4.3 Hyperbolic vision-language models are powerful out-of-distribution discriminators

**Setup.** Lastly, we evaluate the out-of-distribution discrimination potential of hyperbolic image-text embeddings. The ability to discriminate between in- and out-of-distribution samples is a requirement for real-world deployment. Here, we leverage the comprehensive out-of-distribution detection benchmark of Yang et al. (2022). We evaluate three common out-of-distribution detection methods that can be directly applied as a post-processing technique on pre-trained models:

- **MLS**: The Maximum Logit Score (MLS) (Hendrycks et al., 2022) uses the maximum logit. Operating on logits provides compatibility across manifolds and robustness to overconfidence in extreme multi-class scenarios. For hyperbolic CLIP, logits are computed via the Lorentzian inner product (Equation 1) between image embeddings and embeddings of prompted class names: "this is a photo of a [class name]". Then the prediction and confidence score are calculated with $\max_k(\langle \mathbf{E}_\mathbf{I}^*, \mathbf{E}_{\mathbf{T},k}^* \rangle_\mathcal{L})$. For the Euclidean CLIP, we get the maximum over the cosine similarity scores.

- **EBO**: Energy-Based OOD (EBO) scoring applies an energy function to the logit outputs (Liu et al., 2020b). Energy scores are theoretically aligned with the probability density of the inputs and mitigate overconfidence. Like MLS, this score is manifold agnostic and the logits are calculated in the same manner as for MLS. For the hyperbolic CLIP, the predictions are determined by $\max_k(\text{softmax}(\langle \mathbf{E}_\mathbf{I}^*, \mathbf{E}_{\mathbf{T},k}^* \rangle_\mathcal{L}))$. The confidence score is determined with the energy score and since this is non-probabilistic, it can be calculated with the `logsumexp` operator by $\tau \log \sum_k \exp(\langle \mathbf{E}_\mathbf{I}^*, \mathbf{E}_{\mathbf{T},k}^* \rangle_\mathcal{L}/\tau)$, where $\tau$ is the temperature hyperparameter. For the Euclidean CLIP, we get the same expressions by using the cosine similarity score.

- **MDS**: The Mahalanobis Distance Score (MDS) fits the class-conditional Gaussian distribution on the penultimate layer features of the classifier and computes OOD scores using the Mahalanobis distance metric, which is only defined for Euclidean spaces (Lee et al., 2018). For valid comparison, hyperbolic embeddings are first projected back into the Euclidean geometry via the logarithmic map (Equation 8).

  Following Lee et al. (2018), we compute the empirical class mean $\widehat{\mu}_c$ and covariance $\widehat{\Sigma}$ of the training samples. Then the confidence score $M(\mathbf{x})$ is defined by using the Mahalanobis distance between the image embedding $\mathbf{E}_\mathbf{I}$ and the closest class-conditional Gaussian distribution, *i.e.*,

$$M(\mathbf{E_I}) = \max_c \left\{ -(\mathbf{E_I} - \widehat{\mu}_c)^\top \widehat{\Sigma}^{-1} (\mathbf{E_I} - \widehat{\mu}_c) \right\}. \tag{11}$$

  Since MDS operates on the penultimate layer features, similar to the unimodal scenario, we use the image embeddings as an input for the confidence scoring function. For the predictions, we calculate the maximum over the logits which are calculated by the cosine similarity score. Since we chose a specific point of projection from the hyperboloid back to the tangent space, all computations happen in the Euclidean space and are well-defined for the MDS calculation.

Table 6: **Out-of-distribution detection** with the post-processing techniques MLS, EBO, and MDS on CIFAR-10, ImageNet200, and ImageNet-1k by using the AUROC score. For all methods, the improvement by the hyperbolic model is statistically significant with $p < 0.025$, indicated with ‡.

| Method | Model | CIFAR-10 | | ImageNet-200 | | ImageNet-1K | | ImageNet-200 - fsood | | ImageNet-1K - fsood | |
|---|---|---|---|---|---|---|---|---|---|---|---|
| | | Near-OOD | Far-OOD | Near-OOD | Far-OOD | Near-OOD | Far-OOD | Near-OOD | Far-OOD | Near-OOD | Far-OOD |
| MLS | Euclidean CLIP | $84.6 \pm 0.2$ | $90.2 \pm 0.1$ | $79.3 \pm 0.2$ | $87.7 \pm 0.2$ | $63.7 \pm 0.2$ | $74.5 \pm 0.2$ | $68.6 \pm 0.2$ | $79.3 \pm 0.1$ | $60.2 \pm 0.2$ | $71.6 \pm 0.2$ |
| | Hyperbolic CLIP | $\mathbf{86.4 \pm 0.2^{\ddagger}}$ | $\mathbf{93.7 \pm 0.1^{\ddagger}}$ | $\mathbf{80.3 \pm 0.2^{\ddagger}}$ | $\mathbf{88.8 \pm 0.2^{\ddagger}}$ | $\mathbf{66.0 \pm 0.2^{\ddagger}}$ | $\mathbf{75.6 \pm 0.2^{\ddagger}}$ | $\mathbf{69.7 \pm 0.2^{\ddagger}}$ | $\mathbf{80.7 \pm 0.1^{\ddagger}}$ | $\mathbf{63.0 \pm 0.2^{\ddagger}}$ | $\mathbf{73.2 \pm 0.2^{\ddagger}}$ |
| EBO | Euclidean CLIP | $72.8 \pm 0.3$ | $73.2 \pm 0.2$ | $59.7 \pm 0.3$ | $48.8 \pm 0.2$ | $61.0 \pm 0.2$ | $48.3 \pm 0.2$ | $73.7 \pm 0.2$ | $63.1 \pm 0.2$ | $68.6 \pm 0.2$ | $56.3 \pm 0.2$ |
| | Hyperbolic CLIP | $\mathbf{75.1 \pm 0.2^{\ddagger}}$ | $\mathbf{75.1 \pm 0.1^{\ddagger}}$ | $\mathbf{60.3 \pm 0.3^{\ddagger}}$ | $\mathbf{51.3 \pm 0.2^{\ddagger}}$ | $\mathbf{61.2 \pm 0.2^{\ddagger}}$ | $\mathbf{50.0 \pm 0.2^{\ddagger}}$ | $\mathbf{75.6 \pm 0.2^{\ddagger}}$ | $\mathbf{66.8 \pm 0.2^{\ddagger}}$ | $\mathbf{69.4 \pm 0.2^{\ddagger}}$ | $\mathbf{58.5 \pm 0.2^{\ddagger}}$ |
| MDS | Euclidean CLIP | $87.4 \pm 0.2$ | $90.4 \pm 0.1$ | $78.3 \pm 0.2$ | $88.1 \pm 0.1$ | $66.8 \pm 0.2$ | $81.4 \pm 0.1$ | $43.4 \pm 0.2$ | $59.2 \pm 0.2$ | $50.4 \pm 0.2$ | $66.9 \pm 0.1$ |
| | Hyperbolic CLIP | $\mathbf{88.8 \pm 0.2^{\ddagger}}$ | $\mathbf{93.2 \pm 0.1^{\ddagger}}$ | $\mathbf{82.2 \pm 0.2^{\ddagger}}$ | $\mathbf{93.4 \pm 0.1^{\ddagger}}$ | $\mathbf{72.0 \pm 0.2^{\ddagger}}$ | $\mathbf{89.6 \pm 0.1^{\ddagger}}$ | $\mathbf{45.7 \pm 0.2^{\ddagger}}$ | $\mathbf{67.0 \pm 0.2^{\ddagger}}$ | $\mathbf{53.5 \pm 0.2^{\ddagger}}$ | $\mathbf{74.9 \pm 0.1^{\ddagger}}$ |

For these three out-of-distribution techniques, we use the following settings for evaluation:

- **Out-of-distribution type**: we consider two out-of-distribution detection scenarios. 1) Standard out-of-distribution detection considers distribution shifts in semantic content. 2) Full Spectrum out-of-distribution detection additionally takes the covariate shift into account, which in this setup specifically addresses image corruptions, style change, and resampling bias.

- **Metrics**: We use four metrics: 1) false positive rate at 95% true negative rate (FPR@95), 2) the area under the receiver operating characteristic (AUROC) which indicates the probability that the detector correctly separates in- and out-of-distribution samples, and 3) area under the Precision-Recall curve (AUPR) where either ID images are specified as positive (AUPR-in) or 4) out-of-distribution images are seen as positive (AUPR-out) (Yu & Aizawa, 2019).

- **Datasets**: As In-Distribution (ID) datasets, we use CIFAR-10, ImageNet200, and ImageNet-1k. The out-of-distribution datasets are divided into two categories: near-OOD (hard-OOD) and far-OOD (easy-OOD), based on image semantics and empirical difficulty. For CIFAR-10, we use CIFAR-100 and TinyImageNet (Le & Yang) as the near-OOD datasets and MNIST (LeCun et al., 1998), SVHN (Netzer et al., 2011), Textures (Cimpoi et al., 2014), and Places365 (Zhou et al., 2017) as far-OOD datasets. For ImageNet200 and ImageNet we use the following datasets. For near-OOD, we use SSB-hard (Vaze et al., 2022), a dataset composed of images from ImageNet-21K and NINCO by Bitterwolf et al. (2023). For far-OOD, we use iNaturalist (Horn et al., 2018), Textures (Cimpoi et al., 2014), and OpenImage-O (Wang et al., 2022).

For more details regarding the evaluation setup, we refer to Zhang et al. (2023).

**Results.** Table 6 shows AUROC scores for the three OOD methods on near- and far-OOD data. Across all techniques, hyperbolic embeddings consistently improve OOD detection over Euclidean counterparts in both standard and full spectrum settings. The Mahalanobis Distance Score (MDS) performs best overall, achieving a mean score of 72.0% and 89.6% AUROC for near- and far-OOD with hyperbolic embeddings for ImageNet-1K. These gains are significant compared to 66.8% and 81.4% for Euclidean MDS. Interestingly, this improvement comes despite projecting hyperbolic embeddings back into the Euclidean space required for computing Mahalanobis distance. The robustness to this transformation indicates the hyperbolic geometry provides intrinsic distribution-discriminating properties beyond the distance metric alone.

Table 7 breaks down ImageNet1k results by dataset for Mahalanobis scoring, providing per-distribution FPR95, AUROC AUPR-in, and AUPR-out. Significant OOD detection improvements from hyperbolic over Euclidean are observed consistently. These granular outcomes support the aggregate near/far-OOD conclusions from Table 6: hyperbolic geometry is better in out-of-distribution discrimination. The substantial gains demonstrate real-world viability for deploying hyperbolic vision-language models in open-domain applications.

Table 8 presents a comparison of the large, base and small backbones on AUROC scores on ImageNet-1k for all 5 OOD datasets. Where the hyperbolic CLIP outperforms the Euclidean CLIP on the base and large backbones, the Euclidean CLIP performs better for the small backbone. Overall the hyperbolic CLIP base model performs best out of all models.

Table 7: **Out-of-distribution detection** with the MDS method on ImageNet-1k with SSB-Hard and NINCO as near out-of-distribution datasets and iNaturalist, Textures and Openimage-O as far out-of-distribution datasets. Across different near and far and out-of-distribution datasets, hyperbolic CLIP is more robust than its Euclidean counterpart. For all OOD-datasets the improvement by the hyperbolic model is statistically significant with $p < 0.025$, indicated with ‡.

| Dataset | Model | ImageNet-1k | | | |
|---|---|---|---|---|---|
| | | FPR95 ↓ | AUROC ↑ | AUPR-in ↑ | AUPR-out ↑ |
| SSB-hard | Euclidean CLIP | $85.5 \pm 0.3$ | $62.6 \pm 0.2$ | $61.4 \pm 0.3$ | $61.4 \pm 0.2$ |
| | Hyperbolic CLIP | $\mathbf{80.8 \pm 0.3^{\ddagger}}$ | $\mathbf{66.8 \pm 0.2^{\ddagger}}$ | $\mathbf{66.0 \pm 0.2^{\ddagger}}$ | $\mathbf{66.1 \pm 0.2^{\ddagger}}$ |
| NINCO | Euclidean CLIP | $75.5 \pm 0.7$ | $71.0 \pm 0.3$ | $94.8 \pm 0.1$ | $21.3 \pm 0.4$ |
| | Hyperbolic CLIP | $\mathbf{70.2 \pm 1.1^{\ddagger}}$ | $\mathbf{77.3 \pm 0.3^{\ddagger}}$ | $\mathbf{96.0 \pm 0.1^{\ddagger}}$ | $\mathbf{30.4 \pm 0.6^{\ddagger}}$ |
| iNaturalist | Euclidean CLIP | $62.8 \pm 0.5$ | $71.5 \pm 0.2$ | $92.7 \pm 0.1$ | $27.6 \pm 0.3$ |
| | Hyperbolic CLIP | $\mathbf{45.6 \pm 0.6^{\ddagger}}$ | $\mathbf{84.4 \pm 0.2^{\ddagger}}$ | $\mathbf{96.2 \pm 0.1^{\ddagger}}$ | $\mathbf{46.2 \pm 0.5^{\ddagger}}$ |
| Textures | Euclidean CLIP | $41.7 \pm 1.1$ | $90.3 \pm 0.2$ | $98.7 \pm 0.0$ | $60.1 \pm 0.7$ |
| | Hyperbolic CLIP | $\mathbf{26.9 \pm 0.9^{\ddagger}}$ | $\mathbf{94.4 \pm 0.2^{\ddagger}}$ | $\mathbf{99.3 \pm 0.0^{\ddagger}}$ | $\mathbf{73.7 \pm 0.6^{\ddagger}}$ |
| Openimage-O | Euclidean CLIP | $52.7 \pm 0.6$ | $82.4 \pm 0.2$ | $93.2 \pm 0.1$ | $58.1 \pm 0.4$ |
| | Hyperbolic CLIP | $\mathbf{38.5 \pm 0.6^{\ddagger}}$ | $\mathbf{89.9 \pm 0.1^{\ddagger}}$ | $\mathbf{96.2 \pm 0.1^{\ddagger}}$ | $\mathbf{74.5 \pm 0.4^{\ddagger}}$ |

Table 8: **Out-of-distribution detection** on ImageNet-1k with SSB-Hard and NINCO as near out-of-distribution datasets and iNaturalist, Textures and Openimage-O as far out-of-distribution datasets. Results are reported on small, base and large ViT backbones. Across different near and far and out-of-distribution datasets, hyperbolic ViTs are more robust than their Euclidean counterparts for the base and large models. These results are statistically significant with $p < 0.025$, indicated with ‡.

| Dataset | Model | ImageNet-1k | | |
|---|---|---|---|---|
| | | AUROC ↑ | | |
| | | S | B | L |
| SSB-hard | Euclidean CLIP | $64.7 \pm 0.2$ | $65.2 \pm 0.2$ | $62.6 \pm 0.2$ |
| | Hyperbolic CLIP | $62.3 \pm 0.2$ | $\mathbf{68.0 \pm 0.2^{\ddagger}}$ | $66.8 \pm 0.2^{\ddagger}$ |
| NINCO | Euclidean CLIP | $75.1 \pm 0.4$ | $73.9 \pm 0.3$ | $71.0 \pm 0.3$ |
| | Hyperbolic CLIP | $72.7 \pm 0.4$ | $\mathbf{78.0 \pm 0.3^{\ddagger}}$ | $77.3 \pm 0.3^{\ddagger}$ |
| iNaturalist | Euclidean CLIP | $88.8 \pm 0.2$ | $83.3 \pm 0.2$ | $71.5 \pm 0.2$ |
| | Hyperbolic CLIP | $83.7 \pm 0.2$ | $\mathbf{90.2 \pm 0.1^{\ddagger}}$ | $84.4 \pm 0.2^{\ddagger}$ |
| Textures | Euclidean CLIP | $94.5 \pm 0.1$ | $95.0 \pm 0.1$ | $90.3 \pm 0.2$ |
| | Hyperbolic CLIP | $91.8 \pm 0.2$ | $\mathbf{97.3 \pm 0.1^{\ddagger}}$ | $94.4 \pm 0.2^{\ddagger}$ |
| Openimage-O | Euclidean CLIP | $89.4 \pm 0.1$ | $86.9 \pm 0.2$ | $82.4 \pm 0.2$ |
| | Hyperbolic CLIP | $85.8 \pm 0.2$ | $\mathbf{92.2 \pm 0.1^{\ddagger}}$ | $89.9 \pm 0.1^{\ddagger}$ |

**Analysis.** To investigate the OOD gains, Figure 5 plots distribution shifts for Mahalanobis scoring on near-OOD NINCO (left) and far-OOD iNaturalist (right). ID scores (orange) separate further from OOD (blue) with hyperbolic embeddings. The peaks flatten and variance increases, reducing overlap.

Figure 6 shows per-class FPR95 difference between hyperbolic and Euclidean CLIP embeddings on the NINCO near-OOD dataset. Images in NINCo's 64 fine-grained categories are visually similar to some ImageNet classes but belong to different distributions. According to Bitterwolf et al. (2023), images from the NINCO dataset from for instance the categories chicken quesadilla, spaghetti bolognese, and donut are most confusing to ImageNet classes burrito, carbonara, and bagel respectively. Remarkably, hyperbolic embeddings improve OOD detection over Euclidean on 54 out of 64 NINCO categories. Over half exhibit substantial gains above 10% FPR95. This granular analysis confirms and provides insight into the aggregated near-OOD results. There are 9 categories for which the Euclidean CLIP performs 1-6% better than the hyperbolic CLIP, however the absolute performance of the models for these categories varies heavily. 7 out of 9 of these categories are classes from the Species dataset. We did not find a strong correlation between specific characteristics in properties of these classes and their images. Looking at real-world applications, OOD detection is useful for many such as finding unidentified objects in forensics data or finding anomalies in medical imaging data.

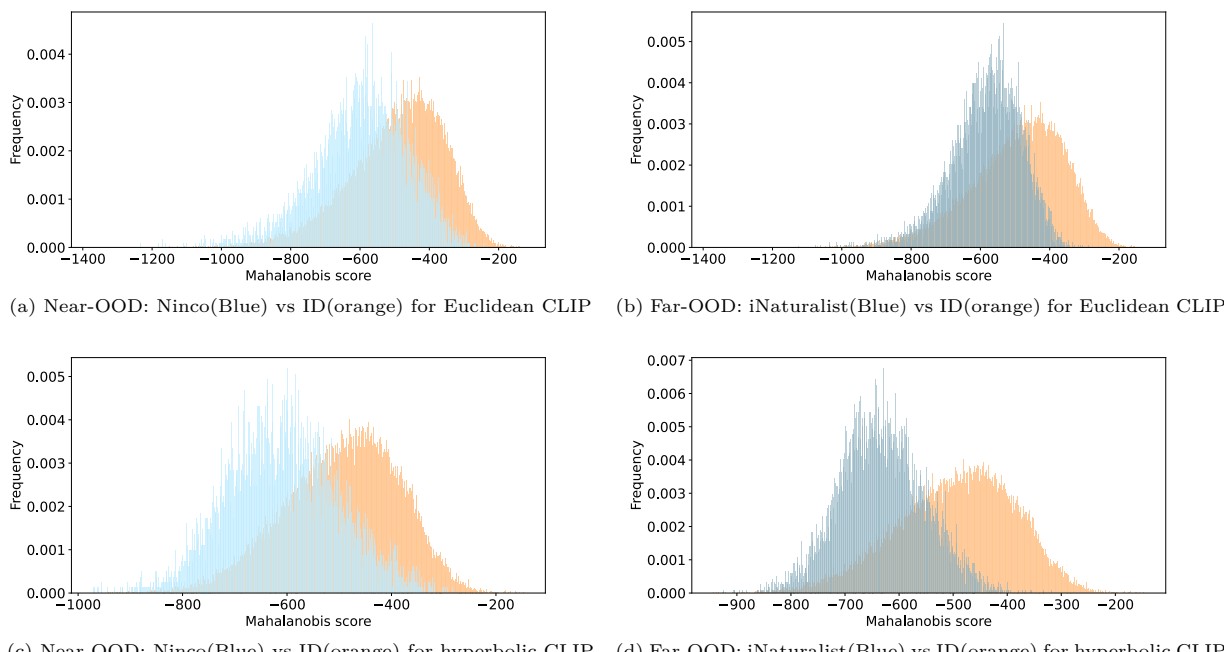

(a) Near-OOD: Ninco(Blue) vs ID(orange) for Euclidean CLIP

(b) Far-OOD: iNaturalist(Blue) vs ID(orange) for Euclidean CLIP

(c) Near-OOD: Ninco(Blue) vs ID(orange) for hyperbolic CLIP

(d) Far-OOD: iNaturalist(Blue) vs ID(orange) for hyperbolic CLIP

Figure 5: **Out-of-distribution detection**. This distribution plot presents the Mahalanobis score (x-axis) and frequency (y-axis) for Euclidean CLIP (top) and hyperbolic CLIP (bottom). The ID and OOD samples show less overlap in the plots for hyperbolic CLIP, which indicates better OOD detection performance.

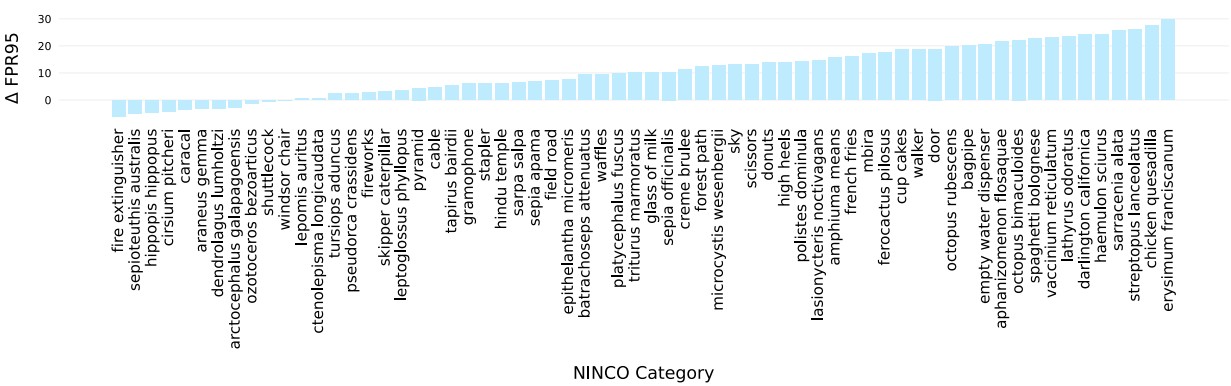

Figure 6: **Out-of-distribution detection** results for NINCO per-class: FPR95 difference between hyperbolic and Euclidean CLIP embeddings. A positive score means that hyperbolic embeddings are better for the considered category.

# 5 Conclusion

Vision-language models have gained a lot of traction due to their broad applicability and strong performance on established down-stream tasks. In vision-language models, the de facto shared embedding space is Euclidean. Recent work shows that this default choice is not the only option and hyperbolic geometry provides a viable alternative with strong performance on often studied benchmarks such as retrieval and zero-shot recognition. In this work, we broaden the scope of the potential of hyperbolic vision-language models and demonstrate their power on their important but under-studied goals.

We find that hyperbolic vision-language models obtain strong spatial reasoning results compared to their Euclidean counterpart, confirming higher intrinsic awareness of visual compositionality. The consistent im-

provements over the Euclidean baseline indicates room for further advances, for instance, by developing specialized fine-tuning strategies targeting compositional tasks. Beyond explicit spatial relationships, analogous ideas may generalize to other latent forms of visual and linguistic compositionality. An investigation of the inductive biases introduced by the hyperbolic geometry remains an intriguing direction for future work. Additionally, hyperbolic models exhibit enhanced reasoning amid ambiguity in text, images, and video. We argued a possible connection between improved ambiguity handling and the hierarchical structure of the embedding space, but further in-depth analysis is required to confirm this. Lastly, hyperbolic vision-language models outperform Euclidean vision-language models in out-of-distribution detection, which complements existing evidence of this advantage in unimodal settings. However majority of the OOD detection techniques require specific fine-tuning and this remains an open challenge for future work.

While hyperbolic vision-language models provide advantages, they also come with challenges. First, hyperbolic operations increase the computational cost of a neural network due to additional nonlinear operations. Compared to the baseline Euclidean model, hyperbolic image-text models include an additional exponential map, while distance calculations are performed with a Lorentzian inner product instead of Euclidean distance. This causes, however, a minimal increase for the running time of our experiments (evaluated on VL-Checklist taking 114.8 seconds for hyperbolic CLIP evaluation and 113.6 seconds for Euclidean CLIP). This is because the main computational load is given by the image and text encoders, which are identical for both models. While a minor increase for evaluation, some additional run time is to be expected during pre-training or fine-tuning due to the use of an additional entailment loss. Current models focus on hyperbolic embeddings only. If transformer layers would be replaced with hyperbolic alternatives as well, it could increase the computational cost drastically and different optimization techniques should be performed to overcome this issue. This is an active area of research studied by Lou et al. (2020), Shimizu et al. (2021), and van Spengler et al. (2023). Another challenge is the numerical stability in the training process that comes from exponential volume growth (Mishne et al., 2023). Where the Poincaré model faces difficulties when dealing with small numbers, the Lorentzian model suffers from dealing with large numbers. These challenges have to be taken into account when developing new models and training them end-to-end.

Overall, to make more progress in this field, fine-tuning hyperbolic vision-language models on more tasks and exposing such models to more data for pre-training are promising next steps. This could be done by exploring new constraints for the fine-tuning process, for example through new loss functions to complement the contrastive and entailment losses, or even different architectural modifications such as the use of hyperbolic layers. A deeper study into the behavior of different model capacity sizes would also be an interesting direction for future research. While the large model gives the most stable performances across all tasks, an in-depth analysis of the structures of the embedding spaces of the different models might provide more insights in their behavior. Due to its potential, we hope that this work can serve as a starting point for more hyperbolic vision-language related research in both the hyperbolic learning and vision-language communities.

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
