# OpenReview forum: "Intriguing Properties of Hyperbolic Embeddings in Vision-Language Models"
_TMLR — Accepted by TMLR_

### Review · Reviewer_fEQu · 2024-03-07

**Summary Of Contributions:**

This paper compares CLIP embeddings to Hyperbolic CLIP embeddings on 3 different axes: spatial awareness, ambiguity resolution and out-of-distribution discrimination.

This is a strictly evaluative paper, which reuses models trained and released by Desai et al 2023 (if I understood correctly, see weakness and requested change below).

Overall I find that the first and third explorations are well executed and show interesting observations that would interest some members of the TMLR community (albeit a small minority). However, the second exploration is more problematic to me, as I find it does not provide sufficient evidence to the claim of “hyperbolic vision-language models are better in handling ambiguity”, in particular due to how they chose to assess ambiguity with the datasets selected.

As it stands, I’m leaning toward rejection, given this issue and the limited scope (even given TMLR’s broad acceptance guidelines).

**Audience:**

Yes

**Claims And Evidence:**

No

**Requested Changes:**

Critical:
1. Clarify model checkpoint usage and on strength of the base CLIP model
2. Comment on the closeness of the scores throughout the paper, how much better would one need to be to be confident about one model performing significantly better than the other?
3. Comment on Exploration 2 and how they provide enough evidence for the current claim.
   1. Alternatively, reduce the claim to a lower level or remove this section.

Strengthen:
1. Comment on other issues raised above.

**Strengths And Weaknesses:**

1. The paper is clear and presents hyperbolic VLM representations well. It presents the 3 explorations well enough and the Setup-Results-Analysis construction of each section was very constructive and helpful.
   1. However, a lot of the theory and supporting knowledge has to be gleaned from reading the original papers, the summary on page 4 is quite terse.
2. I found it hard to clearly understand that the authors did not train any models of their own, and instead just used publicly released models and checkpoints.
   1. This is mentioned in Section 3.5, but I feel like this should be made more prominent.
   2. It is hard to know if the results would have followed when using a reproduced Euclidian CLIP? All numbers are very close to each other, which might just indicate that these particular checkpoints aren’t particularly strong or that the RedCaps dataset isn’t sufficiently complex.
   3. This isn’t discussed in the paper, and the models are taken as is and compared directly. Unless these are standard models that have been reused by the community, this feels like a source of uncertainty about the reproducibility of these observations.
3. Exploration 1 on spatial abilities is well executed. Datasets are chosen appropriately, they target known issues with CLIP and they assess the correct metric.
   1. Figure 3 is interesting, and provides good evidence for the benefit of hyperbolic representations.
   2. However I feel like the results are only slightly better than Euclidean CLIP, and hence the claim of superiority is a bit stretched.
4. Exploration 3 on OOD is also well executed, and shows the strongest and clearest results of the paper. I am not an expert in this domain so would leave others to comment on this, but this section is strong enough for me.
5. Exploration 2 on ambiguity is not as well-defined as the other two, and I am not convinced it provides evidence for the claims made.
   1. It is hard to appropriately quantify how well models handle “ambiguity”, especially given these are contrastive models with no intrinsic measure of uncertainty. I find it strange that the authors consider that doing better on “ambiguous vision-language matching datasets” would imply “handle semantic ambiguity better”.
   2. There are too many alternative failure modes possible that could explain these results without ambiguity or hyperbolic representations providing a clear explanation for the change in metrics.
   3. Table 2 also shows extremely close results for the two models, I would probably not make strong claims based on these.
   4. Only the Visual Word Sense Disambiguation dataset feels like it enables some sort of “hierarchy of semantic specificity”, which I see how this would be enabled by hyperbolic VLMs, but again scores in Table 4 are very close together.
6. Mostly a nitpick, but I find the phrasing of the “hypotheses” to be too opinionated and not actually taking the form of an hypothesis.
   1. For example, hypothesis 3: “hyperbolic VLM are good OOD discriminators”, this is an observation/result, not a hypothesis. The hypothesis ought to be the specific reason/mechanism for why that would hold, which is mentioned in the paragraph below.
   2. Similarly, “Hyperbolic VLMs exhibit spatial awareness” is a strong claim, I do not believe the results warrant such a claim, they are slightly better than Euclidian CLIP at best

---

> ### Author Response · Authors · 2024-04-23
> **Response to Reviewer fEQu**
>
> We thank the reviewer for the feedback and suggestions to improve the paper. We have addressed the raised points below.
>
> # Limited scope of the paper
>
> While hyperbolic learning in vision-language literature is new, image-text models form an actively researched topic in the field. A growing body of research has uncovered that image-text models with a Euclidean embedding space come with blindspots and limitations, ranging from compositionality (Yüksekgönül et al 2023) to polysemy (Raganato et al. 2023).
>
> With this paper, we want to provide in-depth results and analyses showing that the choice of manifold of the shared embedding space forms an important aspect, building upon recent advances in hyperbolic image-text models (Desai et al. 2023). Reviewer KqHF endorses the relevance of this work by emphasizing the broad interest to the community as a strength of the paper. We believe that this paper is of relevance to the entire vision-language community and of interest for many in the representation learning community, not just the hyperbolic learning community.
>
> # Requested change 1: Clarify model checkpoint usage and on strength of the base CLIP model
>
> We apologize for the confusion. The models that have been used in our work are the official checkpoints released by Desai et al. (2023). These are the first hyperbolic vision-language models and to the best of our knowledge currently only available models and therefore set the standard. We extended the motivation for using these models in Section 3.5. Regarding the generalization of the results to the official Euclidean CLIP, Desai et al. (2023) also mention the difficulty of predicting how both the Euclidean and hyperbolic models will behave when trained on a much larger scale. The findings of our paper highlight that already at the current scale, intriguing new properties arise in hyperbolic image-text models compared to the Euclidean standard and we consider larger-scale investigations an interesting future direction.
>
> # Requested change 2: Closeness of the scores and significance
>
> We agree with the reviewer on the importance of an analysis on the closeness and significance of results. We used a bootstrapping technique and statistical significance tests to give more clarity, the details are provided in the general comment to all reviewers in this rebuttal. We show that for almost all experiments the hyperbolic model performs better than the Euclidean model and that results are statistically significant with p<0.025.
>
> # Requested change 3: Claim on Ambiguity
>
> We understand that ambiguity is a word with different meanings in different contexts that should be clarified. We defined the use of ambiguity in the paper:  *“Ambiguity in a machine learning context can be related to uncertainty and to semantics. In this work, we refer to a specific type of semantic ambiguity. The interpretation of memes and other multimodal polysemic challenges involves a grounding problem where the semantics of the text data can only be resolved by grounding it to the image or video, therefore we will refer to 'visually-grounded language ambiguity' in the remainder of this paper whenever we use the term ambiguity.”*
>
> # Other aspects
> 1.1 Terse summary on background: the background section on Hyperbolic Geometry (section 3.1) was presented in a short and concise manner and contained only the definitions and equations that were relevant for our experiments. Following the reviewer’s guidance for completeness, we have extended this section.
>
> 3.2 Superiority in results: We toned down the subjective wordings such as superior and significant and only used significance when the results are in fact statistically significant.
>
> 6.1 The use of ‘hypotheses’: We thank the reviewer for the suggestion and have replaced the use of “hypotheses” by “assumptions” in the paper.
>
> 6.2 ‘Spatial awareness’: our claim regarding spatial awareness is not based on the fact that for most tasks the hyperbolic model is a few points better than the Euclidean model but on the fact that model is clearly able to perform better than random, while the Euclidean model is not. The hyperbolic model is still far from solving the spatial reasoning tasks, but has higher potential.

---

### Review · Reviewer_KqHF · 2024-03-09

**Summary Of Contributions:**

This work investigates the importance of embedding space geometry on zero-shot performance of vision-language models.

Specifically, the work compares:

* A Euclidean CLIP Model
* A Hyperbolic CLIP Model

Using text prompts in the corresponding text encoder to perform zero-shot classification. The Hyperbolic CLIP Model is the MERU type from https://arxiv.org/abs/2304.09172, using a Lorentzian hyperboloid. Both models are taken from https://arxiv.org/abs/2304.09172, are equally sized (ViT-L/16 Image encoder), were trained on the same data (12M image-text pairs from RedCaps https://arxiv.org/abs/2111.11431), and are not further trained in this work.

The work compares the impact of embedding geometry upon model behavior for three types of task through zero shot evaluation:

* Capturing spatial awareness
* Dealing with ambiguity
* Identifying OOD samples


The authors find that:

* Hyperbolic embedding geometry gives consistent performance improvements in all three task types.
* Hyperbolic embedding geometry produces hierarchical embeddings in vision-language models.

**Audience:**

Yes

**Broader Impact Concerns:**

No broader impact concerns.

**Claims And Evidence:**

Yes

**Requested Changes:**

# Required (critical)

* Check statistical significance of results in Table 3. (e.g. bootstrap CI/permutation test)


# Recommended for strengthening but not critical

* Add statistical significance to results throughout.
* Include results with hyperbolic CLIP models pretrained without entailment loss.
* Include results for small medium and models.
* Use vector graphics for all figures where possible.
* (page 1, paragraph 1, line 8) Remove “vastly” (avoid subjective), potentially qualify instead with the nature of the difference.
* (page 1, paragraph 2, final line) Define “common tasks”.
* (page 2, Ambiguity Resolution) Define “distortion”.
* (page 2, Image-Text Representation Leaning) add discussion regarding the origin of vision-language models as potentially from data, not specifically the vision-language aspect (https://arxiv.org/abs/2205.01397), as far as I understand, the origin of CLIP’s robustness being due to multi-modality is now contested
* (page 3, Hyperbolic Learning) Clarify paragraph regarding the limitations of Shen that vision-only hyperbolic models do not perform well, but vision-text hyperbolic models do.
* (page 6, table 1) Explain why worse than random is OK for Euclidean CLIP. I know you mention it in text that this is consistent with findings of other work, it would be useful for readers to see these claims in context here,
* (page 7, propaganda Memes) the description of points 1. - 4. for how embeddings are produced is confusing as the fourth point returns the score itself, not an embedding. Maybe separate point 4. from 1.-3. or, turn 1.-3. into score produces by prepending the inner product with the text embed of prompt class.
* (page 8, My Reaction When) define “common”
* (page 11) Provide equations for MLS and EBO.
* (page 11, MDS) Explain that because we have chosen a specific point of projection from the hyperboloid back to the tangent space, that the inner products are all defined for the MDS calculation (otherwise a reader may think the MDS is doing inner products between vectors existing in different tangent spaces).
* (page 13, Fig 5) Add x and y axes labels, make text larger.
* (page 13, Fig 6) Add error analysis/discussion for elements with negative delta.

**Strengths And Weaknesses:**

# Strengths

The work sets out to measure quantities of broad interest to the community: what are the benefits of geometric spaces other than Euclidean for zero-shot evaluation, and how can we understand any observed performance improvement?

The work is very clearly laid out. The two models chosen for the comparison are as similar as possible, controlling for as many confounders as is realistic.

The conclusions are of high utility, with the performance difference between the two embedding methods being consistent across almost every single evaluation of every task investigated.

# Weaknesses

The work hypothesizes the relation between of latent structure and related performance under ambiguity stemming from the entailment loss of the MERU model. The MERU paper performed ablations investigating the impact of the presence or not of the entailment loss in the pretraining objective. The work would be more informative for the research community if comparisons using a non-entailment pretrained model were included, such that the posed hypothesis is rejected or accepted.

The work only studies models at a single scale (the ViT-L/16 model). Models of medium and small size also available at https://github.com/facebookresearch/meru. It would be more useful to understand also if the trends observed for the large model holds for smaller models.

In result reporting throughout, it is not clear when results are statistically significantly different, as opposed to e.g. the accuracies reported just being larger in one case than the other. This is particularly problematic in Table 3, where many of the numbers from Euclidean and Hyperbolic CLIP are similar (or even the same at this level of numerical significance), and only the Hyperbolic CLIP result are boldened.

There are cases where Hyperbolic CLIP fails compared to Euclidean CLIP (e.g. see Figure 6), the authors do not explain these/provide any error analysis.

The presentation of some of the figures would be improved, e.g. figures should be vector graphics where possible. Figure 5 has no x/y axis levels and the text is small, Figure 6 the text is small.

---

> ### Author Response · Authors · 2024-04-23
> **Response to Reviewer KqHF**
>
> We thank the reviewer for the positive comments on the broad interest for the community, clarity, and high utility of the conclusions. Below we discuss all raised points one-by-one.
>
> # Statistical significance
>
> We thank the reviewer for the suggestion to focus on statistical significance tests. We describe the details of the implemented bootstrapping method and new evaluations in the general rebuttal comment. For Table 3, the results on the validation set of *My Reaction When* are provided below. For results indicated by *, the results of the hyperbolic model being better than the Euclidean model are statistically significant. In the paper, we updated tables 1,2,3,4,6,7.
>
> **Val-set (1k):  Text-Video retrieval**
> |Model|r@1| r@5|r@10 |r@20 |
> | --| --| --| --| --|
> |Euclidean CLIP |0.2 &plusmn; 0.1|0.8 &plusmn; 0.3|1.7 &plusmn; 0.4|3.3 &plusmn; 0.5|
> |Hyperbolic CLIP|**0.2 &plusmn; 0.1** |**1.3 &plusmn; 0.4**\*|**2.4 &plusmn; 0.5**\*|**4.2 &plusmn; 0.6**\*|
>
> **Val-set (1k):  Video-Text retrieval**
> |Model|r@1| r@5|r@10 |r@20 |
> | --| --| --| --| --|
> |Euclidean CLIP|0.2 &plusmn; 0.1|0.6 &plusmn; 0.2|1.6 &plusmn; 0.4	|2.9 &plusmn; 0.5|
> |Hyperbolic CLIP|**0.5 &plusmn; 0.2**\*|**1.2 &plusmn; 0.4**\*|**2.4 &plusmn; 0.5**\*|**4.2 &plusmn; 0.6**\*|
>
> 13 out of 16 results are statistically significant. More details of this analysis can be found in Table 3 in the paper.
>
> # Results on medium and small models
>
> For each of the three properties, we added results for the medium (ViT-B/16) and small (ViT-S/16) models. These results can be found in Tables 1, 4, and 8. A summary for each property is provided here:
> - Spatial Awareness: for VL-Checklist and CLIPbind-r, the large hyperbolic model performs best out of all model sizes. In particular, for CLIPbind-R the large hyperbolic model is the only model that performs better than random. For VG-Relations, the small model performs best out of all sizes, with the hyperbolic model being better than the Euclidean model. Overall, the large hyperbolic model has the most stable performance across all tasks.
> - Ambiguity: we evaluated the model sizes for the Visual Word Sense Disambiguation task. For all metrics, the large hyperbolic model is the best model.
> - Out-of-distribution detection: for the large and base model, the hyperbolic model performs best. For the small model, the Euclidean model performs better. Overall, the medium hyperbolic model performs best for both near-OOD and far-OOD datasets.
>
> We can conclude that the large model is the best performing model, especially because for spatial awareness it performs better than random on 2 benchmarks. While the large model gives the most stable performances across all tasks, an in-depth analysis of the structures of the embedding spaces of the different models might provide more insights in their behavior.
>
> # Entailment loss
> The open-source release of the hyperbolic CLIP model MERU (Desai et al. 2023) only includes the Euclidean baseline and the hyperbolic model with entailment loss. We have asked the authors to also share the hyperbolic baseline without entailment loss, but we were unfortunately not given this model. From Table 4 in (Desai et al. 2023), we find that the effect of the entailment loss itself is minimal. We see the strength of the hyperbolic model in all its components and have emphasized this in the paper by removing attributions of specific contributions to the entailment loss.
>
> # Explaining failure cases
> We have added the following about the failure cases of the NINCO dataset from Figure 6: *“There are 9 categories for which the Euclidean CLIP performs 1-6% better than the hyperbolic CLIP, however the absolute performance of the models for these categories varies heavily. 7 out of 9 of these categories are classes from the Species dataset. We did not find a strong correlation between specific characteristics in properties of these classes and their images.”*
>
> # Clarification on worse than random scores for CLIPbind-r
> We want to clarify that we do not believe that worse than random is ok for any of the models. It is a phenomenon that seems to happen with more models due to the nature of the dataset and therefore we believe that it is interesting that the hyperbolic model in fact performs better than random, and it is worth studying why the hyperbolic model is better on this task for future research. We believe that hyperbolic embeddings provide a more fruitful basis for future studies on spatial aware reasoning in vision-language models.
>
> We thank the reviewer for the detailed list of other recommendations. They have all been addressed in the text and figures throughout the paper and can be found in the revision uploaded version of the paper.

---

> > ### Comment · Reviewer_KqHF · 2024-05-02
> > **Comment on response**
> >
> > I thank the authors for their detailed response and updates to the paper. The readability of the paper is much improved. Introducing the other two model sizes, as well including statistical significance testing using resampling has increased the the work's utility for the research community.
> >
> > The provided results for the OOD testing as requested by reviewer Grex are also interesting and a welcome inclusion.
> >
> > One minor thing I noticed when reading the updated manuscript that I recommend amending:
> > - Equation (9): change $sinh\rightarrow \sinh$ to maintain style of e.g. equation (8)

---

### Review · Reviewer_Grex · 2024-04-15

**Summary Of Contributions:**

This paper offers a compelling analysis of the advantages of using hyperbolic embeddings within the context of vision-language (VL) models. While researchers have touched on the benefits of hyperbolic embeddings, this work goes a step further by focusing specifically on large-scale VL settings.  The authors uncover several key properties that showcase the unique advantages of hyperbolic embeddings.

Firstly, the paper demonstrates that VL models using hyperbolic embeddings develop an intrinsic spatial awareness that models based on traditional Euclidean embeddings do not possess. This translates to a marked improvement in tasks that demand a nuanced understanding of spatial relations between objects or concepts. Secondly, compared to their Euclidean counterparts, hyperbolic VL models are more adept at handling ambiguity present in both language and visual modalities. This likely stems from the inherent ability of hyperbolic space to accommodate hierarchical structures and represent multiple interpretations simultaneously.  Lastly, the authors show that hyperbolic VL models excel at distinguishing between differing distributions of data, a characteristic that could pave the way for greater robustness in out-of-distribution generalization and the detection of anomalies in real-world applications.

Collectively, these findings shed new light on the reasons behind the performance gains observed when hyperbolic embeddings are employed in VL models, extending the discussion beyond simple metrics and towards a broader potential for impact.

**Audience:**

Yes

**Claims And Evidence:**

Yes

**Requested Changes:**

Computational Cost Analysis: Providing an analysis of the computational complexity of the proposed hyperbolic VL models in comparison to their Euclidean equivalents is essential. If a significant increase in cost exists, suggesting potential optimization techniques or outlining research directions to address this would be crucial for the practical adoption of these methods.

**Strengths And Weaknesses:**

Strengths:

* Novelty: The exploration of spatial awareness, ambiguity handling, and distribution discrimination specifically within hyperbolic VL models is a significant new contribution to the field.
* Experimental Rigor: The experiments seem well-designed, and the results support the authors' claims about the advantages of hyperbolic embeddings.
* Clarity and Organization: The paper is clearly written, flows logically, and makes effective use of examples and figures to illustrate complex concepts.

Weaknesses
* Narrower Dataset Focus: The authors could broaden the types of datasets used to further validate the universality of their findings.
* Unexplored Limitations: A deeper discussion of potential shortcomings of hyperbolic embeddings (e.g., computational costs or challenges in fine-tuning) would strengthen the work.
* Real-World Use Cases: Providing more concrete examples of real-world scenarios where the highlighted properties of hyperbolic VL models translate into tangible benefits would bolster the paper's impact.

---

> ### Author Response · Authors · 2024-04-23
> **Response to Reviewer Grex**
>
> We thank the reviewer for the positive feedback on the novelty, rigor, and organization of the paper. Below, we address the raised points.
>
> # Computational Cost Analysis
> We agree with the reviewer that this is an important aspect of a complete analysis and added the following paragraph to the Conclusion section of the paper:
>
> *“Hyperbolic operations increase the computational cost of a neural network due to additional nonlinear operations. Compared to the baseline Euclidean model, hyperbolic image-text models include an additional exponential map, while distance calculations are performed with a Lorentzian inner product instead of Euclidean distance. This causes a minimal increase for the running time of our experiments (evaluated on VL-Checklist taking 114.8 seconds for hyperbolic CLIP evaluation and 113.6 seconds for Euclidean CLIP). This is because the main computational load is given by the image and text encoders, which are identical for both models. While a minor increase for evaluation, some additional run time is to be expected during pre-training or fine-tuning due to the use of an additional entailment loss. Current models focus on hyperbolic embeddings only. If transformer layers would be replaced with hyperbolic alternatives as well, it could increase the computational cost drastically and different optimization techniques should be performed to overcome this issue. This is an active area of research studied by Lou et al. (2020), Shimizu et al. (2021) and Van Spengler et al. (2023).”*
>
> # Broadening datasets
> The first (spatial awareness) and second (ambiguity resolution) properties that we evaluate in this work are relatively new research areas in the vision-language domain and have therefore a limited number of available benchmarks. To the best of our knowledge, we use all datasets that are available for these properties.
>
> For out-of-distribution detection, we provide results on additional datasets following the OpenOOD benchmark of Yang et al. (2022). More specifically, we use CIFAR-10 as the ID dataset and this comes with a new set of OOD datasets, namely CIFAR-100, TinyImageNet, MNIST, SVHN, Textures and Places365. We use the AUROC score for evaluation. The results are presented below and in Table 6 of the paper.
>
> |Method |Model | Near-OOD | Far-OOD |
> | -- | ---- | ---- | ---- |
> |MLS | Euclidean CLIP | 84.6 &plusmn; 0.2| 90.2 &plusmn; 0.1 |
> | |Hyperbolic CLIP|**86.4 &plusmn; 0.2**\*|**93.7 &plusmn; 0.1**\*|
> |EBO| Euclidean CLIP|72.8 &plusmn; 0.3|73.2 &plusmn; 0.2|
> ||Hyperbolic CLIP|**75.1 &plusmn; 0.2**\*|**75.1 &plusmn; 0.1**\*|
> |MDS|Euclidean CLIP|87.4 &plusmn; 0.2|90.4 &plusmn; 0.1|
> ||Hyperbolic CLIP|**88.8 &plusmn; 0.2**\*|**93.2 &plusmn; 0.1**\*|
>
> The new results are consistent with the previous out-of-distribution results, with hyperbolic CLIP outperforming the Euclidean alternative. Furthermore, to deepen the empirical evaluations on all the properties, we have performed bootstrap experiments to extract deeper statistical understanding of our performance compared to the Euclidean baseline, with new results for Tables 1,2,3,4,6,7,8.
>
> # Unexplored limitations
> Following the guidance of the reviewer, besides the computational cost, we described a second limitation related to numerical instability in the conclusion section of the paper:
>
> *“Another challenge is the numerical stability in the training process that comes from exponential volume growth (Mishne et al. 2023) . Where the Poincaré model faces difficulties when dealing with small numbers, the Lorentzian model suffers from dealing with large numbers. These challenges have to be taken into account when developing new models and training them end-to-end.”*
>
> # Real-World Use Cases
> The three properties that we studied in this work, spatial awareness, ambiguity, and out-of-distribution detection, are all very important aspects when using vision-language models for real-world scenarios. One should be careful with deploying such models in high impact systems, because mistakes with severe consequences could be made. For each of the three properties we added more concrete examples of real-world use cases where that specific property is of high importance. These examples are added to the ‘Analysis’ subsections of Section 4 on Results.

---

### Author Response · Authors · 2024-04-23
**General comments and uploaded revision of the paper**

We thank the reviewers for their feedback and suggestions to improve the paper. A revised version of the paper has been uploaded, with all changes highlighted in blue. We will address all reviewer comments in the individual responses. Below, we address the most important reviewer comments and the updates to the paper.

# Evaluating statistical significance

We have included statistical analyses throughout our manuscript as requested by Reviewer **KqHF** and **fEQu**, and we follow the recommended setup from Reviewer **KqHF**. The details are added to Section 3.5 of the paper and are as follows:

*“Since we have a single checkpoint for each model, averaging the results over multiple checkpoints is not possible. However, we provide statistical significance tests on the results by using a bootstrapping technique on the test set. More specifically, we follow Sanchez-Lengeling et al. (2019) by using a resampling method with replacement on the test set for 500 times, where the number of times is selected based on common practice. This results in a distribution of 500 data points for which we report the mean and standard deviation for all experiments. We perform one-sided t-tests on the data distributions of the Euclidean and hyperbolic models with the null hypothesis stating that the **data distribution of the hyperbolic model has a higher mean value than the Euclidean model** with a significance level of 2.5%. Statistically significant results are indicated with \*.”*

We show that on all tasks the hyperbolic model with the ViT-L/16 backbone outperforms the Euclidean model and that these results are mostly statistically significant with p<0.025. This is also the case for the ‘My Reaction When’ dataset that was in particular mentioned by Reviewers **fEQu** and **KqHF**, for which 13 out of 16 results are statistically significant, even though the results are close together.

# Additional experiments

We have performed the following additional experiments based on the reviewers’ feedback:
- The bootstrapping experiments with statistical significance tests that are explained above (reviewers **KqHF** and **fEQu**). This resulted in extensions of Tables 1,2,3,4,6,7,8.
- Evaluations on more datasets (reviewer **Grex**): for OOD we add experiments on the new ID dataset CIFAR-10 and this comes with a new set of OOD datasets, namely CIFAR-100, TinyImageNet, MNIST, SVHN, Textures, and Places365. We evaluate the MLS, EBO and MDS methods on these datasets. The results are presented in Table 6.
- Evaluation on base (ViT-B/16) and small (ViT-S/16) model sizes (reviewer **KqHF**): this is done for the three properties on spatial awareness, ambiguity, and out-of-distribution detection. This results in extensions of Tables 1 and 4 and the new Table 8.

We provide more details in the individual responses and in the new pdf version of the paper.

# Typos and recommended renaming

We thank the reviewers for their guidance regarding the writing. We have addressed the other concerns to increase readability and clarity. Details are given in the individual responses.

---

### Decision · Action_Editor_hVcE · 2024-06-13

**Recommendation:** Accept as is

**Comment:**

Seems like good empirical work, of interest to a wide range of folks!

**Audience:**

Pretty much anyone doing research with vision-language models should find this work useful and interesting (that should be a good chunk of the TLMR readership).

**Claims And Evidence:**

This work analyses using hyperbolic embeddings in vision-language models, especially in large-scale settings. The work contrasts them with Euclidean embeddings and elucidates several advantages of hyperbolic ones. Notably: spatial understanding, handling ambiguity and distinguishing between different distributions of data.

In general, the work presents a novel study of these embeddings with good experiments and it is well-written. The authors have accommodated a lot of the reviewers' requests and that process has significantly strengthened the claims made in the paper.